# Impaired $\alpha$-Synuclein aggregate clearance in neuronal cells drive their spread to microglia through tunneling nanotubes

Ranabir Chakraborty[1,2], Francesca Palese[1], Philippa Samella[1,3], Veronica Testa [4,7], Jara Montero-Muñoz [4,7], Sylvie Syan[1], Takashi Nonaka[5], Masato Hasegawa [5], Antonella Consiglio [4] & Chiara Zurzolo [1,6] ✉

Tunneling nanotubes (TNTs) play a crucial role in intercellular communication, enabling transfer of molecular cargoes over long distances between connected cells. Previous studies have demonstrated efficient, directional transfer of $\alpha$-Synuclein ($\alpha$-Syn) aggregates from neurons to microglia, with endosomal trafficking and lysosomal processing identified as the primary events following $\alpha$-Syn internalization. Using human neuronal and microglial cell lines, we show that microglia exhibit higher lysosomal turnover, particularly through lysophagy, whereas neuronal lysosomes display compromised degradative capacity and impaired autophagic flux upon $\alpha$-Syn exposure, resulting in compromised aggregate clearance. Such a response to $\alpha$-Syn aggregates is also conserved in human iPSC-derived neurons and microglia. Moreover, perturbing aggregate clearance via autophagy inhibition enhances TNT-mediated transfer of $\alpha$-Syn from neuronal cells to microglia. Microglia co-cultured with $\alpha$-Syn-containing neurons upregulate autophagy flux, enabling efficient degradation of the transferred aggregates. These results highlight dysfunctional autophagy in neurons as a key driver outsourcing $\alpha$-Syn aggregates to microglia.

Parkinson's Disease (PD) is the second most common neurodegenerative disease, the worldwide burden of which has doubled between 1990 and 2016[1]. One of the major genetic associations of sporadic PD, and the first to be discovered, is *SNCA*, which encodes a cytosolic protein $\alpha$-Synuclein ($\alpha$-Syn). Both mutations in *SNCA*, as well as elevated levels of the protein (due to duplication and triplication of the gene), can give rise to a pathological state[2–4].

A common pathological hallmark of several neurodegenerative diseases, including PD, is disrupted proteostasis. Under physiological conditions, quality control pathways in cells allow for efficient clearing of misfolded proteins, thereby preventing their accumulation and/or

aggregation[5]. Autophagy, a key proteostasis pathway, orchestrates the degradation of obsolete intracellular components, including organelles and aggregated proteins, via lysosomal machinery. Several reports highlight compromised macroautophagy (henceforth referred to as autophagy) or chaperone-mediated autophagy (CMA) in PD: impaired autophagosome formation[6], dysregulated mitophagy due to mutations in genes encoding PTEN-induced putative kinase 1 (*PINK1*) and Parkin RBR E3 ubiquitin-protein ligase (*PARKIN*)[7,8], decreased CMA caused by mutations in leucine-rich repeat kinase 2 (*LRRK2*) or *SNCA*[9,10], and impaired lysosomal activity due to mutations in the glucocerebrosidase-encoding gene *GBA1*[11]. Besides genetic mutations,

[1]Institut Pasteur, CNRS UMR 3691, Université Paris Cité, Paris, France. [2]Université Paris-Saclay, Gif-sur-Yvette, France. [3]Trinity College, University of Cambridge, Cambridge, UK. [4]Department of Pathology and Experimental Therapeutics, Bellvitge University Hospital-IDIBELL, 08908 Hospitalet de Llobregat, Spain, Institute of Biomedicine of the University of Barcelona (IBUB), Carrer Baldiri Reixac 15-21, Barcelona, Spain. [5]Dementia Research Project, Tokyo Metropolitan Institute of Medical Science, Tokyo, Japan. [6]Department of Molecular Medicine and Medical Biotechnology, University of Naples Federico II, Naples, Italy. [7]These authors contributed equally: Veronica Testa, Jara Montero-Muñoz. ✉e-mail: chiara.zurzolo@pasteur.fr

in vitro overexpression of α-Syn in human mesencephalic neuronal (LUHMES) cells has also been reported to impair autophagy by preventing the fusion of autophagosomes with lysosomes[12]. Additionally, overexpression of α-Syn in human neuroblastoma (SH-SY5Y) cells impairs starvation-induced autophagy[13]. Such dysregulation prompts cells to seek alternative mechanisms to restore homeostasis, failure of which may culminate in apoptosis. Of note, the accumulation of intracellular α-Syn aggregates resulting from dysregulated autophagy has been associated with their secretion via exosomes and membrane shedding[14–17]. Moreover, recent findings elucidate the role of microglia in alleviating neuronal α-Syn burden through a process termed synucleinphagy, wherein microglia engulf aggregates released by neurons and target them towards autophagy-mediated clearance[18]. However, whether α-Syn aggregates have a different impact in lysosomal and autophagic pathways in neuronal versus microglial cells, and whether microglia modulate proteostasis in response to neuronal α-Syn burden, remain unknown.

Since their discovery in 2004[19], tunneling nanotubes (TNTs) have emerged as pivotal conduits for contact-dependent cell-to-cell communication that mediate the exchange of various molecular cargo, from small ions to larger organelles[20]. Moreover, these membranous, F-actin-rich structures facilitate the movement of neurodegenerative disease-associated protein aggregates such as α-Syn, tau, and Amyloid-β[21,22]. Notably, a related structure termed *dendritic nanotubes* (DNTs) was recently identified between dendrites of neurons in the mammalian cortex[23]. In primary neuronal cultures and mouse brain tissue, DNTs were shown to actively transport calcium ions and amyloid-β (Aβ) between connected neurons. α-Syn aggregates exploit TNTs to traverse between diverse cell types, including neuronal precursor cells (NPCs)[24], neurons[25,26], astrocytes[27,28], and microglia[29,30]. Even though previous studies have demonstrated the augmentation of homotypic connections between neuronal cells and microglial cells by α-Syn aggregates[25,30], the mechanism underlying their preferential directional transfer from neuronal cells to microglia remains enigmatic.

In this study, using both human cell lines and (hiPSC)-derived neurons and microglia, we investigate the distinct subcellular dynamics of lysosomal processing abilities, both in the presence and absence of α-Syn aggregates. We also examine the non-cell autonomous influence of aggregate-burdened neuronal cells on autophagy regulation in microglia. Our findings uncover a fundamental imbalance in lysosomal function and autophagic activity between neuronal cells and microglia, resulting in compromised aggregate clearance in both neuronal line and hiPSC-derived neurons. This results in neuronal cells outsourcing α-Syn aggregates via TNTs to microglia for autophagic degradation. They advance our understanding of how organellar dysfunction shapes neuron–microglia crosstalk in the context of pathological protein aggregation.

## Results

### α-Syn localization to lysosomes in neuronal cells and microglia

Exogenous α-Syn has been reported to be internalized by cells via different mechanisms, dependent on dynamin[31], clathrin[32], cell surface receptors such as heparan sulfate proteoglycans (HSPGs)[33], and lymphocyte activation gene 3 (LAG3)[34], as well as macropinocytosis[35]. Internalized α-Syn aggregates eventually reach the lysosomes in different cell types[25,26,31,35]. To assess the extent and kinetics of α-Syn preformed fibrils (PFF) trafficking to lysosomes in neuronal and microglial cells, we challenged both human SH-SY5Y neuronal and HMC3 microglial cells with fluorescently labeled PFFs[36] for different time durations, ranging from 1 to 16 h, and performed immunostaining against lysosome-associated membrane protein 1 (LAMP1) (Fig. 1a). We observed a time-dependent increase in the presence of α-Syn aggregates with lysosomes for both the cell types, however with different kinetics. The extent of aggregates association with lysosomes at the different time points differed markedly between neurons and

microglia (Fig. 1b, c). In neuronal cells, following α-Syn exposure for 16 h, an average of 67.58% of lysosomes overlapped with α-Syn, whereas only 29.30% of microglial lysosomes localized with aggregates at that time point. The increase in lysosomal localization over time was steeper in neuronal cells than in microglia, as reflected by the slope of the linear regression fit (0.04249 vs. 0.01620, respectively). Structured illumination super-resolution microscopy further confirmed greater co-localization of α-Syn aggregates with neuronal lysosomes than with microglial lysosomes after 16 h of exposure (Fig. 1d, e; *cyan asterisks*). Taken together, these results highlight cell–type–specific differences in aggregate–lysosome association, suggesting a potential disparity in the impact of α-Syn aggregates on lysosomal function.

### α-Syn aggregates differentially affect lysosomes in neuronal cells and microglia

To assess the impact of α-Syn aggregates on lysosomal function, we evaluated lysosomal degradative capacity, pH, and motility. Cells were first incubated with Dextran-647 (10 kDa) for 3 h (pulse phase), followed by exposure to α-Syn aggregates for 16 h (chase phase), during which Dextran traffics from early endosomes and predominantly localizes to lysosomes[37]. Two hours before the end of the chase period, cells were incubated with dye-quenched bovine serum albumin (DQ-BSA), which is internalized and trafficked to lysosomes (Fig. 2a). Catalytic cleavage of DQ-BSA-dye results in a substantial increase in fluorescence[38], thereby allowing to probe the functionality of lysosomes.

Exposure to α-Syn aggregates markedly reduced the proportion of degradative lysosomes (DQ-BSA+Dextran+; indicated by cyan asterisks in representative images) per cell in both the cell types (Fig. 2b–d). However, the reduction was more pronounced in neuronal cells—dropping from 81.86% under control conditions to 45.01% following aggregate treatment—compared to a smaller decline in microglia (from 83.59 to 75.63%). These findings were further supported by quantification of DQ-BSA fluorescence intensity within Dextran+ lysosomes, a read-out of lysosomal degradative ability, which revealed a significant reduction in neuronal cells but not in microglia (Fig. 2e, f). Since lysosomal degradative capacity depends on their pH, we also assessed lysosomal acidity using the pH-sensitive dye lysosensor green. In agreement with the degradative lysosome analysis, neuronal lysosomes displayed a more substantial reduction in acidity upon exposure to α-Syn aggregates compared to microglial lysosomes, while such a reduction was observed in both cell types upon bafilomycin A1 treatment (Fig. S1a–c).

We also observed a differential effect on the functionality of lysosomes localized with aggregates (α-Syn+/Dextran+). In neuronal cells, these lysosomes exhibited reduced DQ-BSA fluorescence, indicating compromised degradative function, whereas in microglia, lysosomes with α-Syn aggregates largely retained their functionality (Fig. 2g–i). Taken together, these findings highlight a cell type–specific vulnerability, revealing that neuronal lysosomes are more functionally sensitive to α-Syn aggregates at the single-organelle level.

Lysosomal functionality depends on their membrane integrity. Lysosome membrane permeabilization (LMP) has been reported in both PD and cells exposed to extracellular α-Syn aggregates[25,39,40]. To prevent such ruptures, different ESCRT proteins are known to be recruited on damaged lysosomes to facilitate membrane repair[41]. To examine the occurrence of LMP and lysosomal membrane repair in neuronal and microglial cells upon α-Syn treatment, we performed immunostaining against Galectin-3, a marker of endo-lysosomal rupture (Fig. S1d), and the ESCRT-III associated factor IST-1 (Fig. S1e). We observed increased LMP upon aggregate exposure in both neuronal cells compared to microglial cells (Fig. S1f, g). However, the extent of LMP was more than 3-fold higher in neuronal cells compared to microglia [2.83% (of total lysosomes) in neuronal cells versus 0.78% in

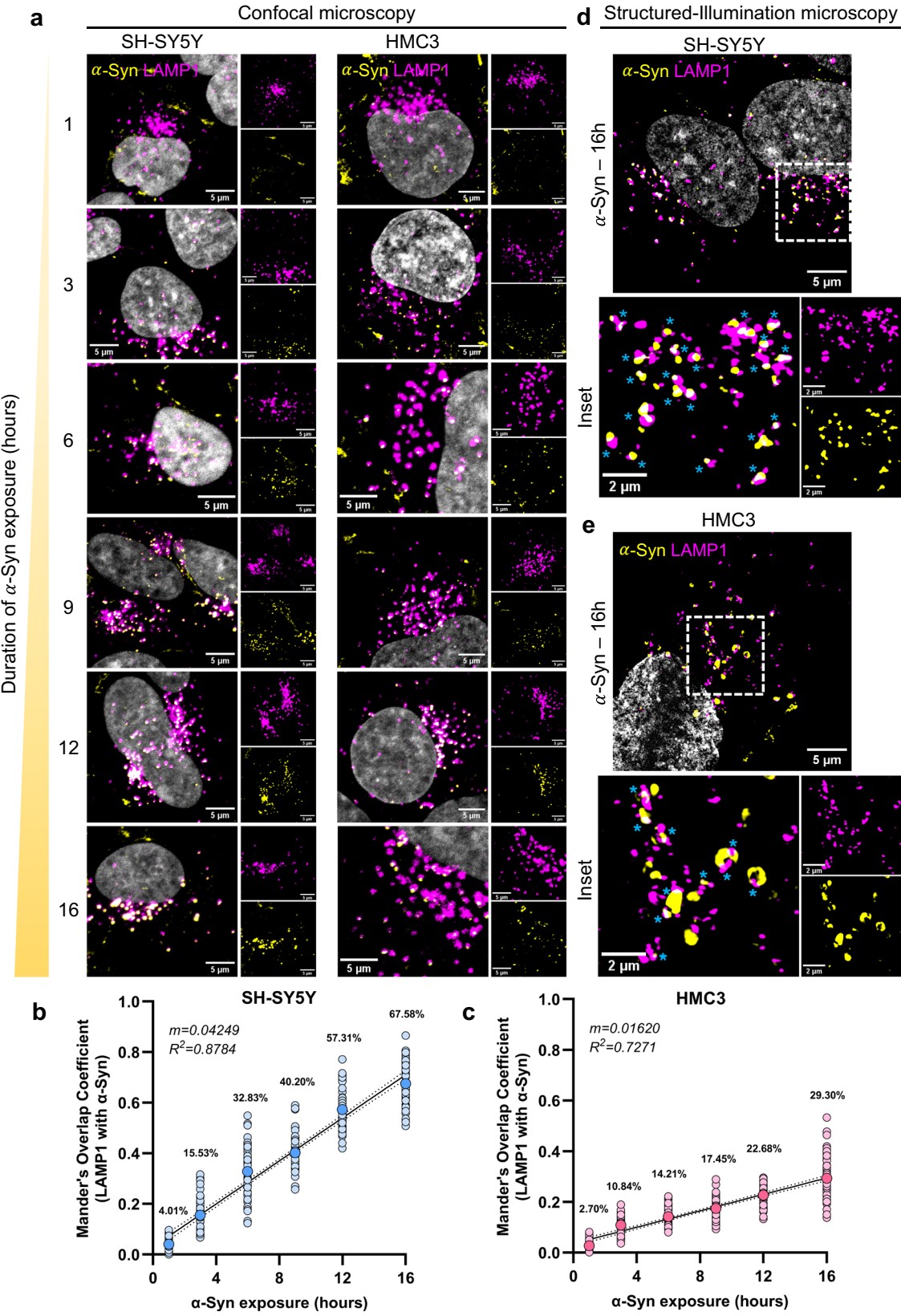

microglia]. Accordingly, the need for membrane repair was also different between the two cell types (2.82% of neuronal lysosomes being IST-1+, versus 1.13% for microglia) (Fig. S1h, i). Taken together, these results demonstrate higher sensitivity of neuronal lysosomes to α-Syn aggregates compared to those in microglia, highlighting the relative resilience of the latter.

Given the importance of lysosomal trafficking in regulating their positioning, and consequently their function[42], we assessed whether aggregate association influenced lysosomal motility. Using lysotracker green, we tracked individual lysosomes over 2 min in control cells and in cells treated with aggregates for 16 h, comparing lysosomes containing aggregates (+α-Syn) or not (-α-Syn) (Fig. S2a–d; and

**Fig. 1 | Temporal analysis α-Syn localization in neuronal cells and microglia.**
**a** Representative confocal images of α-Syn (pseudo-colored yellow) localization with lysosomes (pseudo-colored magenta) at different (indicated) time points of aggregate exposure of SH-SY5Y neuronal cells (left panels) and HMC3 microglia (right panels). **b, c** Quantification of co-localization between lysosomes and aggregates (Mander's overlap coefficient−fraction of LAMP1 with α-Syn) at different time points of aggregate exposure of neuronal cells (**b**) and microglia (**c**). N = 3 independent experiment, n = 50 cells per group. Larger, darker filled circle depicts the mean, also mentioned as percentages within the graphs. Black line denotes the simple linear regression fit; dotted lines represent the 95% confidence intervals. Structured-Illumination microscopy images of neuronal cells (**d**) and microglia (**e**) exposed to aggregates for 16 h show localization of α-Syn to lysosomes. Images are pseudo-colored−magenta for LAMP1-647, yellow for α-Syn488, and gray for DAPI. Cyan asterisks denote areas of α-Syn−lysosome colocalization. Source data are provided as a Source data file.

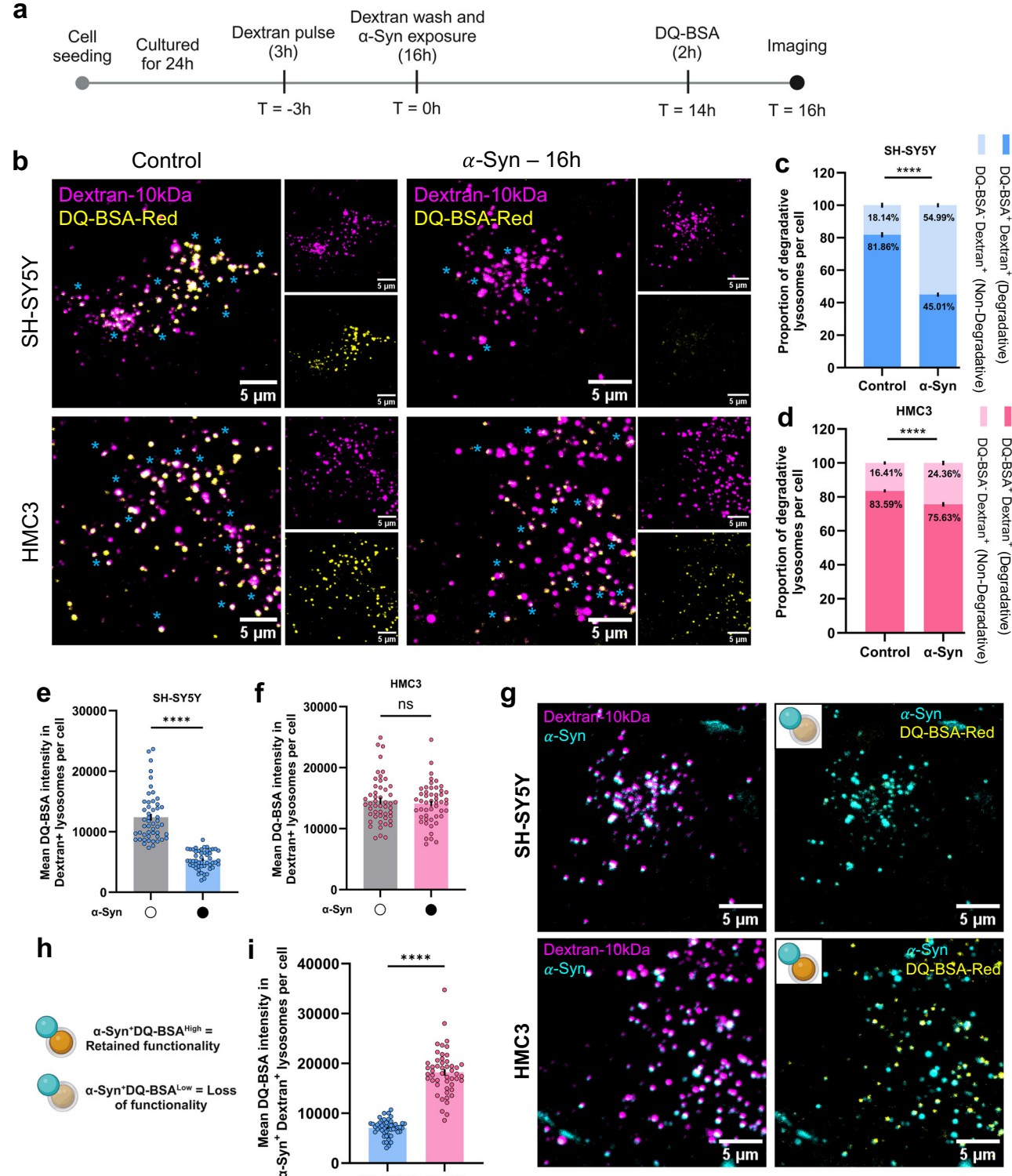

**Fig. 2 | Effect of $\alpha$-Syn aggregates on lysosomal degradative ability. a** Schematic representation of the experimental design to assess for degradative lysosomes. **b** Representative images of neuronal and microglial cells pulse-chased for Dextran647-10 kDa (magenta) and DQ-BSA red (yellow) in SH-SY5Y (upper panels) and HMC3 (lower panels) cells. Quantification of the proportion of degradative lysosomes per cell as a percentage of total number of lysosomes for neuronal cells (**c**) and microglia (**d**). $N = 3$ independent experiments, $n = 50$ cells per group. Mean values are mentioned within the graph, error bars represent SEM. Statistical significance was analyzed using Two-Way ANOVA with uncorrected Fisher's LSD multiple comparison. ****$p < 1 \times 10^{-15}$ (DQ-BSA⁻Dextran⁺ - Control versus $\alpha$-Syn) for (**c**) and $p = 2.27 \times 10^{-6}$ for (**d**). Quantification of mean DQ-BSA fluorescence

intensities in Dextran+ neuronal (**e**) and microglial (**f**) lysosomes (lysosomal degradative ability per cell). $N = 3$ independent experiments, $n = 50$ cells per group. Statistical significance was analyzed using two-tailed Mann–Whitney U test. ns ($p = 0.8611$), ****$p < 1 \times 10^{-15}$. **g** Representative dual-channel images of neuronal cells (left panels) and microglia (right panels) treated with $\alpha$-Syn488 (cyan). Schematic representation **h** of the quantification **i** of degradative ability of lysosomes associated with $\alpha$-Syn. $N = 3$ independent experiments, $n = 50$ cells per group. Error bars in (**i**) represent SEM. Statistical significance was analyzed using two-tailed Mann-Whitney U test. ****$p < 1 \times 10^{-15}$. **a, h** Created in BioRender. Palese, F. (2026) https://BioRender.com/3zimbkm. Source data are provided as a Source data file.

supplementary movies 1–6). The presence of aggregates on lysosomes significantly reduced both their mean travel distance (Fig. S2e, g) and their mean velocities (Fig. S2f, h) in both neuronal and microglia cells. Additionally, aggregate-bearing lysosomes were more likely to be classified as less motile—defined arbitrarily as traveling ≤15 µm, based on the median track length of control neuronal lysosomes (16.837 µm) —in both cell types (Fig. S2i, j). Notably, the proportion of less motile lysosomes was higher in neuronal cells than in microglia under both control and $\alpha$-Syn–treated conditions (34% vs. 14% in controls, and 54% vs. 30% in aggregate-containing lysosomes). A cumulative frequency distribution further reported a higher number of lysosomes with reduced distance traveled upon association with $\alpha$-Syn aggregates (Fig. S2k, l; arrow depicting a leftward shift in the peak).

Taken together, these data indicate that neuronal lysosomes are more susceptible to the presence of aggregates than those in microglia, resulting in compromised degradative capability. These observations are consistent with the greater localization of aggregates to lysosomes in neuronal cells (Fig. 1), ultimately contributing to the observed reduction in lysosomal functionality.

### Lysophagy and lysosome biogenesis are elevated in microglia
Building on our observation of higher aggregate localization with, and consequent damage to, lysosomes in neuronal cells, we hypothesized that the resilience of microglial lysosomes could be attributed to a more efficient clearance and biogenesis of these organelles in microglia compared to neuronal cells. Since p62 has been reported to be recruited to damaged lysosomes to facilitate their clearance via lysophagy[43], we assessed the extent of colocalization of LAMP1-positive lysosomes with p62 under different conditions. To distinguish lysosomes destined for lysophagy from autolysosomes (which also show p62-LAMP1 co-localization), we conducted this experiment in the presence of bafilomycin A1, which prevents autophagosome-lysosome fusion, thereby allowing the identification of lysosomes marked for lysophagy by p62 (Fig. 3a, cyan dotted circles in inset images). To induce lysosomal damage, L-Leucyl-L-Leucine methyl ester (LLOMe) was used as a positive control. Upon endocytosis, LLOMe is processed within lysosomes to generate a membrane-disrupting polymer, thereby causing LMP. As expected, the extent of lysosomes undergoing lysophagy was significantly increased upon LLOMe exposure in both neuronal and microglial cells (Fig. 3b, c). Similarly, exposure to aggregates for 16 h also increased lysophagy in both cell types. In neuronal cells, lysophagy increased from a basal level of 5.95–21.16% upon aggregate exposure (a 15.21% increase) (Fig. 3b). In microglia, lysophagy rose from a basal level of 25.63–49.20% (a 23.57% increase) (Fig. 3c). Notably, lysophagy was significantly higher in microglia compared to neuronal cells, both at a basal level, and following aggregate exposure (Fig. 3d). These results point towards an enhanced ability of microglia to clear damaged lysosomes and restore proteostasis, a process that appears to be less efficient in neuronal cells.

As part of lysophagic flux, the clearance of organelles necessitates homeostatic compensation through lysosomal biogenesis. Transcription factor EB (TFEB) is a major regulator of autophagy and lysosome-

associated genes[44]. We performed immunostaining against TFEB in both neuronal cells and microglia and assessed the extent of its nuclear translocation. Consistent with our observation of increased lysophagy within 16 h of aggregate exposure, nuclear translocation of TFEB also increased in both the cell types (Fig. 3e–g). At a basal state, TFEB is phosphorylated on Serine 142 and 211 residues by mechanistic target of rapamycin (mTOR), maintaining its cytosolic distribution[45–47]. Immunoblot analysis against Ser211-phosphorylated TFEB revealed a substantial reduction, albeit not statistically significant, in the phosphorylation levels upon $\alpha$-Syn exposure in both the cell types (-34% in neuronal cells, and -29% in microglia), in accordance with TFEB nuclear translocation (Fig. S3a–c). Such a reduction was the result of reduced mTOR activity, as assessed by the reduction in phosphorylation of S6 ribosomal protein, one of the downstream targets of mTOR (Fig. S3d–f).

To further evaluate TFEB activity, we performed RT-PCR to quantify the mRNA levels of TFEB-regulated lysosomal target genes. Upon aggregate exposure, microglial gene expression of the lysosomal protein (*LAMP1*), hydrolases (*CTSD* and *CTSB*), and a subunit of the vacuolar ATPase (*ATP6V1H*) significantly increased relative to control, a phenotype not observed in neuronal cells (Fig. 3h, i). Taken together, these results suggest an increase in lysophagy upon exposure to $\alpha$-Syn aggregates in both the cell types, with microglia being relatively more efficient in lysosomal turnover and biogenesis compared to neuronal cells.

### $\alpha$-Syn aggregates differentially affect autophagic flux in neuronal cells and microglia
To understand the consequence of the differential lysosomal response to $\alpha$-Syn between neuronal cells and microglia, we investigated whether aggregates had any impact on autophagy, given its reliance on functional lysosomes. A robust method for measuring autophagy flux is to assess the conversion of LC3B from its non-lipidated (LC3B-I) form to its lipidated form (LC3B-II), which is associated with autophagosomes (Fig. 4a). As expected, bafilomycin A1 treatment of cells led to an accumulation of LC3B-II in both neuronal and microglial cells (Fig. 4a–c). Interestingly, $\alpha$-Syn exposure of neuronal cells led to a 2-fold reduction in LC3B-II levels compared to control (Fig. 4b). However, in $\alpha$-Syn–exposed neuronal cells treated with bafilomycin A1, LC3B-II levels had a trend to be lower than in cells treated with bafilomycin A1 alone (by -13%), suggesting a possible reduction in autophagy flux. In microglial cells, by contrast, the reduction in LC3B-II levels upon $\alpha$-Syn exposure relative to control was -14%, while co-treatment of $\alpha$-Syn and bafilomycin A1 led to a -9% increase compared to bafilomycin A1 treatment alone, suggesting an unperturbed flux (Fig. 4c). Interestingly, LC3B-II level analysis in high exposure blots (for non-bafilomycin A1 treated conditions) also revealed a stark reduction in neuronal cells exposed to $\alpha$-Syn (-44%), compared to microglial cells (-13%) (Fig. S3g, h). Furthermore, LC3B-I to LC3B-II conversion was also compromised in aggregate-treated neuronal cells, but not in microglial cells, as observed by -27% decline in LC3B-II-to-LC3B-I ratio, although not statistically significant (Fig. S3i, j). Taken together, these results

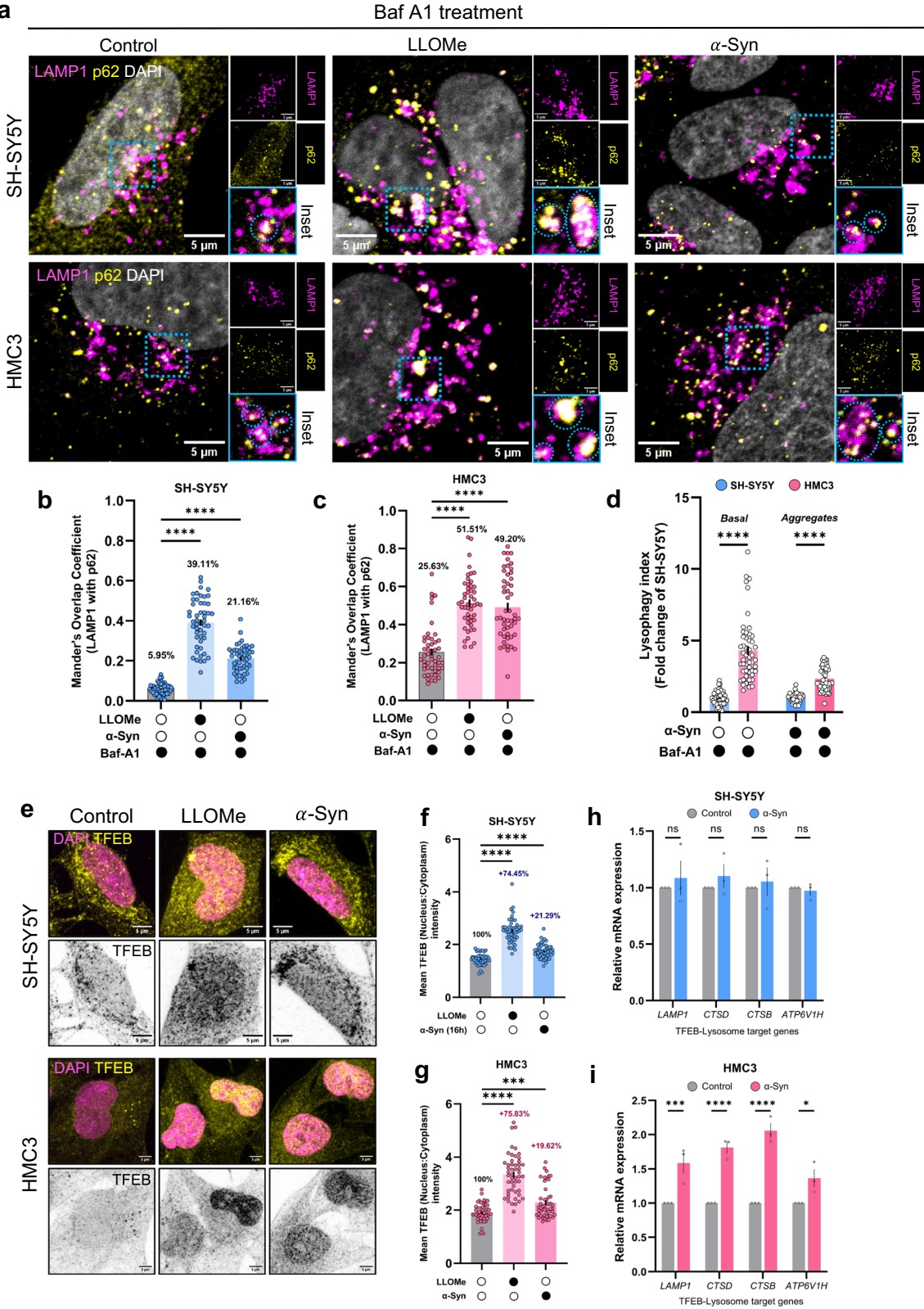

highlight a deficiency in neuronal autophagy in the presence of $\alpha$-Syn aggregates relative to microglia.

The autophagic receptor p62/SQSTM1 (hereafter p62) has been shown to play a central role in aggregate clearance in different neurodegenerative diseases by sequestering cargos in condensates known as p62 bodies, eventually targeting them towards degradation[48–50]. Notably, under our experimental conditions, exposure to $\alpha$-Syn in

both neuronal cells and microglia for 16 h led to a significant increase in the number of p62 puncta per cell (Fig. 4d–f). However, this augmentation could result from either an accumulation of p62 (indicative of autophagy flux dysfunction) or an upregulation of p62 expression (suggestive of autophagy flux upregulation). To discern between these possibilities, aggregate-loaded cells were treated with bafilomycin A1 or left untreated, followed by immunostaining for p62. In neuronal

**Fig. 3 | Effect of aggregates on lysophagy and lysosome biogenesis.**
**a** Representative images of SH-SY5Y and HMC3 cells immunostained for LAMP1 and p62 in the presence of bafilomycin A1. **b** Mander's overlap coefficient (LAMP1 with p62) for SH-SY5Y cells. Mean percentage of co-localization is mentioned within the graph. $N = 3$ independent experiments, $n = 50$ cells per group. Error bars represent SEM. Statistical significance was analyzed using Brown-Forsythe and Welch One-Way ANOVA with Dunnett's T3 multiple comparison. ****$p < 1 \times 10^{-15}$. **c** Same quantification as in (**b**), but in HMC3 cells. Error bars represent SEM. Statistical significance was analyzed using Kruskal–Wallis test with Dunn's multiple comparison. ****$p = 1.32 \times 10^{-12}$ for control versus LLOMe; $p = 3.12 \times 10^{-10}$ for control versus $\alpha$-Syn. **d** Fold-difference in lysophagy induction between neuronal and microglial cells under basal conditions or upon $\alpha$-Syn exposure. $N = 3$ independent experiments, $n = 50$ cells per group. Error bars represent SEM. Statistical significance was analyzed using Two-Way ANOVA with Šídák's multiple comparison. ****$p < 0.0001$. **e** Representative images of TFEB immunostaining (yellow and gray-invert) in SH-SY5Y and HMC3 cells in different conditions. **f** Quantification of TFEB translocation to the nucleus in SH-SY5Y cells. $N = 3$ independent experiments, $n = 50$ cells per

group. Error bars represent SEM. Statistical significance was analyzed using Brown-Forsythe and Welch ANOVA with Dunnett's T3 multiple comparison. Control versus LLOME, $p < 1 \times 10^{-15}$; control versus $\alpha$-Syn, $p = 1.47 \times 10^{-8}$. **g** Same quantification as in (**f**), but in HMC3 cells. $N = 3$ independent experiments, $n = 50$ cells per group. Error bars represent SEM. Statistical significance was analyzed using Brown-Forsythe and Welch ANOVA with Dunnett's T3 multiple comparison. Control versus LLOME, $p < 1 \times 10^{-15}$; control versus $\alpha$-Syn, $p = 8.77 \times 10^{-4}$. **h** Relative expression level of TFEB-lysosomal target genes in SH-SY5Y cells. $N = 3$ biological replicates. Error bars represent SEM. Statistical significance was analyzed using Two-Way ANOVA with Šídák's multiple comparison (comparison between control and $\alpha$-Syn-exposed cells). ns: non-significant ($p = 0.9157$ for *LAMP1*, 0.8492 for *CTSD*, 0.9822 for *CTSB*, and 0.9989 for *ATP6V1H*). **i** Relative expression level of TFEB-lysosomal target genes in HMC3 cells. $N = 3$ biological replicates. Error bars represent SEM. Statistical significance was analyzed using Two-Way ANOVA with Šídák's multiple comparison (comparison between control and $\alpha$-Syn-exposed cells). *$p = 0.0325$, ***$p = 0.0007$, ****$p = 2.05 \times 10^{-5}$ for *CTSD* and $7.13 \times 10^{-7}$ for *CTSB*. Source data are provided as a Source data file.

---

cells, no significant increase in the number of p62 puncta was observed upon bafilomycin A1 treatment in the presence of aggregates (Fig. 4e). In contrast, microglia cells showed a substantial increase in p62 puncta under the same conditions (Fig. 4f). To further validate the status of autophagy flux upon $\alpha$-Syn exposure, we assessed the expression of the autophagy target gene of TFEB, *SQSTM1* (Fig. 4g, h). Consistent with the increase in p62 puncta, we also observed an upregulation of *SQSTM1* expression in microglial cells upon aggregate exposure. Taken together, these results suggest that autophagy flux is impaired in neuronal cells upon $\alpha$-Syn exposure, and not in microglia.

Having observed a difference in autophagy flux between the two cell types in the presence of aggregates, we next investigated whether these aggregates are targeted to lysosomes. To this end, we co-immunostained for LC3 and LAMP1 (Fig. 4i) or p62 and LAMP1 (Fig. 4l) in neuronal cells and microglial cells exposed to $\alpha$-Syn aggregates. We quantified the fraction of cells displaying triple localization events (aggregates co-localizing with an autophagy marker and lysosome), as well as the number of such events per cell. Both autophagy markers participated in triple localization events ($\alpha$-Syn-LC3-LAMP1 and $\alpha$-Syn-p62-LAMP1) in the two cell types. While over 70% of microglia exhibited triple localization events with both LC3 and p62, neuronal cells showed only 26% of triple localization with LC3 and 35.24% with p62 (Fig. 4j, m). In addition, the average number of triple-localized foci per cell was significantly higher in microglia compared to neuronal cells (Fig. 4k, n). In line with our previous observations, these results point towards a more efficient microglial autophagic flux and $\alpha$-Syn aggregates targeting for autophagic degradation compared to neuronal cells.

### Human iPSC-derived neurons and microglia display similar responses to $\alpha$-Syn aggregates as cell lines

To validate our findings in a physiologically more relevant model system, we derived control dopaminergic neurons (DAn) and microglia (MG) from induced pluripotent stem cells (hiPSC) (Fig. S4a, b). DAn and microglia were generated following previously published protocols and characterized using cell-specific markers[51,52]. After 16 h of exposure to $\alpha$-Syn aggregates, a higher proportion of neuronal lysosomes localize with $\alpha$-Syn aggregates compared to microglia (45.45% versus 13.05%) (Fig. 5a–d), consistent with our initial observations in cultured cell lines (Fig. 1).

We attributed the lower percentage of $\alpha$-Syn-LAMP1 colocalization in microglia to reduced number of aggregates in these cells, a factor that critically influences object-based colocalization analyses (i.e., a lower number of aggregates per cell at the time of detection results in a reduced colocalization index). To test this hypothesis, we analyzed the mean fluorescence intensity of $\alpha$-Syn per cell in the

presence or not of bafilomycin A1 (Fig. 5e, f). Not only did we detect a higher level of aggregates upon bafilomycin A1 treatment, suggestive of these aggregates being cargoes of autophagic degradation, but we also detected lesser aggregates in microglia relative to neurons at a basal state (Fig. 5g), suggesting that microglia are relatively efficient in clearing these aggregates, an observation in line with our previous data.

Furthermore, we assessed the impact of $\alpha$-Syn aggregates on autophagy flux in these cells. After 16 h of aggregate exposure, we did not detect any significant increase in the number of p62 puncta per cell relative to control conditions in both neurons (Fig. 5h, i) and microglia (Fig. 5j, k). However, bafilomycin A1 treatment did not increase any further the number of p62 puncta in neurons but did so in microglia. Taken together, these results suggest that $\alpha$-Syn aggregates impair autophagy flux in neurons, but not in microglia, mirroring our previous observations in cell lines (Fig. 4).

### State of neuronal autophagy regulates transfer *of $\alpha$-Syn aggregates to microglia*

Building on our observations and on previous work showing a directional bias in $\alpha$-Syn transfer from neuronal cells to microglia[30], we hypothesized that impaired autophagy in neuronal cells may promote the disposal of $\alpha$-Syn aggregates toward microglia. Consistent with reduced autophagic clearance in neurons, we first found that neuronal cells harbored a higher aggregate load than microglia under basal conditions (Fig. S5a, b). To directly test whether the state of neuronal autophagy influences aggregate transfer, we pharmacologically modulated autophagy in $\alpha$-Syn-burdened cells. On one hand, inhibition of autophagy with wortmannin, an irreversible PI3K inhibitor that blocks autophagosome formation[53], increased $\alpha$-Syn aggregate load both in neurons and in microglia (Fig. S5a, c, d). On the other hand, activation of autophagy with rapamycin reduced aggregate load (Fig. S5e, f), thereby providing a set-up to assess the contribution of autophagy in mediating material transfer between cells.

We then co-cultured aggregate-containing SH-SY5Y neurons (N) pre-treated or not with wortmannin (N^Wm) or rapamycin (N^Rapa) with microglia (M) for 12 h (Fig. 6a, b, f). Wortmannin treatment significantly increased the percentage of microglia receiving aggregates and the number of aggregates per acceptor cell (N^Wm-M vs. N-M; Fig. 6c, d), whereas rapamycin treatment significantly reduced both parameters (N^Rapa-M vs. N-M; Fig. 6g, h). In all conditions, transfer via supernatant remained minimal compared to co-culture, and neither drug significantly altered $\alpha$-Syn secretion (Fig. 6e, i). These results indicate that manipulating neuronal autophagy is sufficient to modulate $\alpha$-Syn transfer to microglia.

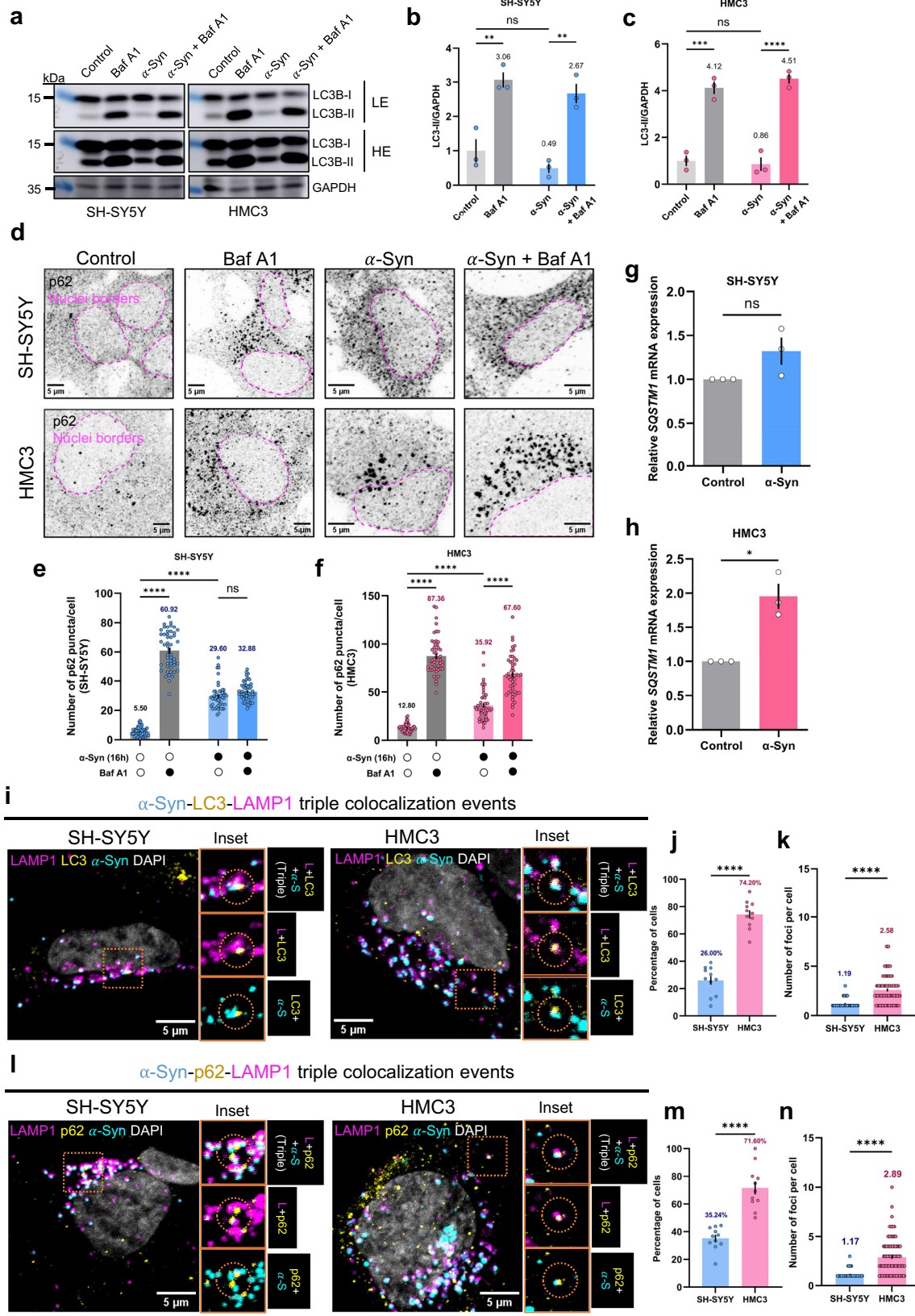

However, these experiments alone cannot distinguish whether the increased transfer arises primarily from greater aggregate load or from autophagy impairment per se. To address this, we next varied the initial $\alpha$-Syn load independently of autophagy status. Neuronal cells were exposed to different $\alpha$-Syn concentrations (100 and 250 nM, in addition to the 500 nM concentration used in previous assays) and

then co-cultured with microglia under basal conditions (N-M) (Fig. S6a). Basal transfer was stable across doses, with ~31–35% of microglia accepting aggregates and ~2.2–2.5 aggregates per acceptor cell (Fig. 6c, d; Fig. S6b–e), indicating that within this range, aggregate load alone does not determine transfer efficiency. In contrast, autophagy inhibition with wortmannin increased aggregate spread to

**Fig. 4 | Effect of aggregates on autophagy flux, and its recognition as a cargo.**
**a** Representative LC3B immunoblots of SH-SY5Y and HMC3 cells. LE low exposure, HE high exposure. Quantification of the relative LC3B-II level in SH-SY5Y (**b**) and HMC3 (**c**) cells. $N = 3$ independent experiments. Error bars represent SEM. Statistical significance was analyzed using Two-Way ANOVA with Tukey's multiple comparison. For (**b**): control versus Baf A1, **$p = 0.0027$; control versus α-Syn, ns ($p = 0.735$); α-Syn versus α-Syn+Baf A1, **$p = 0.0019$. For (**c**): control versus Baf A1, ***$p = 0.0001$; control versus α-Syn, ns ($p = 0.9992$); α-Syn versus α-Syn + Baf A1, ****$p = 2.91 \times 10^{-5}$. **d** p62 immunofluorescence in SH-SY5Y and HMC3 cells. Number of p62 puncta per cell for SH-SY5Y **e** and HMC3 **f** cells. For both (**e**, **f**), $N = 3$, $n = 50$ cells per group. Error bars represent SEM. Statistical significance was analyzed using Two-Way ANOVA with Tukey's multiple comparison. For (**e**): ns ($p = 0.2175$), control versus Baf A1, ****$p = 7.6 \times 10^{-14}$; control versus α-Syn, ****$p = 7.6 \times 10^{-14}$. For (**f**): control versus Baf A1, ****$p = 7.6 \times 10^{-14}$; control versus α-Syn, $p = 1.58 \times 10^{-10}$; α-Syn versus α-Syn+BafA1, $p = 1.13 \times 10^{-13}$. Relative transcript expression of *SQSTM1*

in SH-SY5Y **g** and HMC3 **h** cells. $N = 3$ biological replicates. Error bars represent SEM. Statistical significance was analyzed using two-tailed Student's t-test with Welch's correction. ns ($p = 0.1734$); *$p = 0.0355$. **i** Representative images of triple colocalization of α-Syn with LC3 and lysosomes. **j** Percentage of SH-SY5Y and HMC3 cells with at-least one triple colocalization event. $N = 3$, $n = 212$ SH-SY5Y and 133 HMC3 cells from 11 independent images. ****$p = 7.16 \times 10^{-10}$. **k** Average number of triple-colocalizing foci per cell for SH-SY5Y and HMC3. $N = 3$ independent experiments, $n = 56$ SH-SY5Y and 98 HMC3. ****$p = 6 \times 10^{-15}$. **l** Same as in (**i**), but with p62. **m** Percentage of SH-SY5Y and HMC3 cells with at-least one triple colocalization event. $N = 3$ independent experiments, $n = 201$ SH-SY5Y and 155 HMC3 from 11 independent images. ****$p = 5.23 \times 10^{-6}$. **n** Average number of triple-colocalizing foci per cell for SH-SY5Y and HMC3. $N = 3$ independent experiments, $n = 69$ SH-SY5Y and 109 HMC3. ****$p < 1 \times 10^{-15}$. For (**j**, **k**, **m**, **n**), error bars represent SEM. Statistical significances were analyzed using two-sided unpaired Student's t-test with Welch's correction. Source data are provided as a Source data file.

microglia at all α-Syn concentrations tested (Fig. 6b–d; S6b–e). Similar effects were observed when wortmannin was applied to acceptor microglia (N-M$^{Wm}$) or to both neuronal and microglial populations (N$^{Wm}$-M$^{Wm}$), with the highest aggregate burden in acceptor cells observed when both donor and acceptor were treated (Fig. S7a–c). Together with the load-titration data, these findings show that the enhancement in aggregate transfer is not simply a function of initial α-Syn burden but reflects a key contribution of autophagy status in both donor and acceptor cells. Collectively, these results highlight neuronal autophagy as a critical regulator of α-Syn aggregate spread to microglia, with autophagy inhibition promoting, and autophagy activation limiting, contact-dependent transfer, beyond the effect of aggregate load alone.

As a proof-of-concept that autophagy impairment leads to increased aggregate transfer, we performed a similar experiment, but with microglia serving as the donor cells (Fig. S7d). Consistent with our previous results[30], in control conditions, we observed less transfer of aggregates from microglia to neuronal cells (4.80% in M-N group), which increased to 13.34% upon autophagy impairment (M$^{Wm}$-N) (Fig. S7e, f).

**Compromise of autophagy promotes formation of functional intercellular connections.** Since wortmannin treatment did not increase aggregate release into the medium (Fig. 6e), the enhanced transfer observed in co-cultures was likely mediated by TNTs[30]. We therefore quantified TNT-connected cells under conditions of autophagy inhibition. Wortmannin treatment increased the proportion of TNT-connected cells by 1.83-fold in SH-SY5Y neurons and 1.67-fold in HMC3 microglia (Fig. S8a, b). To assess whether these additional connections were functionally competent, we evaluated the transfer of DiD-labeled vesicles from neuronal cells to microglia (Fig. S8c). DiD labels intracellular vesicles and provides a general read-out of TNT-mediated material exchange. Consistent with the increase in percentage of connections, wortmannin-treated neuronal cells transferred significantly more DiD$^+$ vesicles to microglia than untreated controls (Fig. S8d, e), with no corresponding increase in vesicle release into the medium (Fig. S8f, g). Thus, autophagy inhibition enhances the formation of functional intercellular connections. To independently validate this finding using a mechanistically distinct autophagy inhibitor, we treated cells with bafilomycin A1, which blocks autophagosome–lysosome fusion. Bafilomycin A1 also caused a >1.5-fold increase in intercellular connections in both neuronal cells and microglia (Fig. S8h, i), supporting a general role for autophagy impairment in promoting TNT formation.

The formation of TNTs depends on dynamic actin remodeling. We therefore first examined whether autophagy inhibition affects global actin organization. Quantification of actin filament coherency[54] revealed a significant increase in directional actin alignment in both

neuronal and microglial cells following wortmannin or bafilomycin A1 treatment (Fig. S9a–d), consistent with conditions favoring protrusion formation. We next investigated whether actin regulators previously implicated TNTs formation[55,56] were affected by autophagy inhibition. Immunoblotting for ARP3 and its upstream regulator CDC42 showed a reduction in both proteins in neuronal cells following autophagy inhibition, although the decrease in CDC42 did not reach statistical significance. In contrast, no such changes were detected in microglia (Fig. S9e–g). Together, these observations suggest that autophagy impairment in neuronal cells selectively alters cytoskeletal remodeling in a manner that may support the formation of intercellular connections.

Autophagy inhibition is known to increase oxidative stress, and ROS has been implicated in TNT induction. To test whether elevated ROS might explain the increase in TNT-mediated transfer, we quantified ROS levels in neuronal cells and microglia treated with α-Syn alone or in combination with wortmannin or bafilomycin A1 (Fig. S10a–d). While α-Syn treatment alone induced ROS accumulation (Fig. S10b, d), neither autophagy inhibitor increased ROS beyond α-Syn-induced levels. Thus, although ROS may contribute to baseline TNT formation under α-Syn exposure, it does not account for the additional increase in transfer observed upon autophagy inhibition.

## Neuronal cells with α-Syn induce autophagy in bystander microglia

Cells grown in co-cultures often behave differently compared to those grown in monocultures. The phenomenon of bystander influence on autophagy has been observed during irradiation, known as the radiation-induced bystander effect (RIBE)[57,58]. Additionally, chemokines secreted by activated microglia have been reported to inhibit neuronal autophagy in neurodegenerative pathologies[59]. To investigate potential non-cell autonomous effects in our system, we co-cultured microglia with neuronal cells loaded or not with α-Syn aggregates and performed immunostaining against p62. Autophagy flux was assessed by treating co-cultures with bafilomycin A1 (Fig. 7a, b). We observed a significant increase in the number of p62 puncta per microglial cell when cultured with aggregate-burdened neuronal cells (Nα$^{-Syn}$ + M; average of 27.26; 1.92-fold increase), compared to co-cultures with healthy neuronal cells (N$^{WT}$ + M; average of 14.20) (Fig. 7c). This increase was further enhanced by bafilomycin A1 treatment, indicating upregulated autophagic flux. To assess whether this effect was only contact-dependent, we performed a similar experiment on monocultures of microglial cells exposed to conditioned medium from either naïve or α-Syn-laden neuronal cells (Fig. 7d). We observed a similar increase in the number of p62 puncta per cell when exposed to conditioned medium from aggregate-treated neuronal cells (Nα$^{-Syn}$; average of 22.00; 2.01-fold increase) relative to that from healthy cells (N$^{WT}$; average of 10.92) (Fig. 7e, f). Taken together, these results

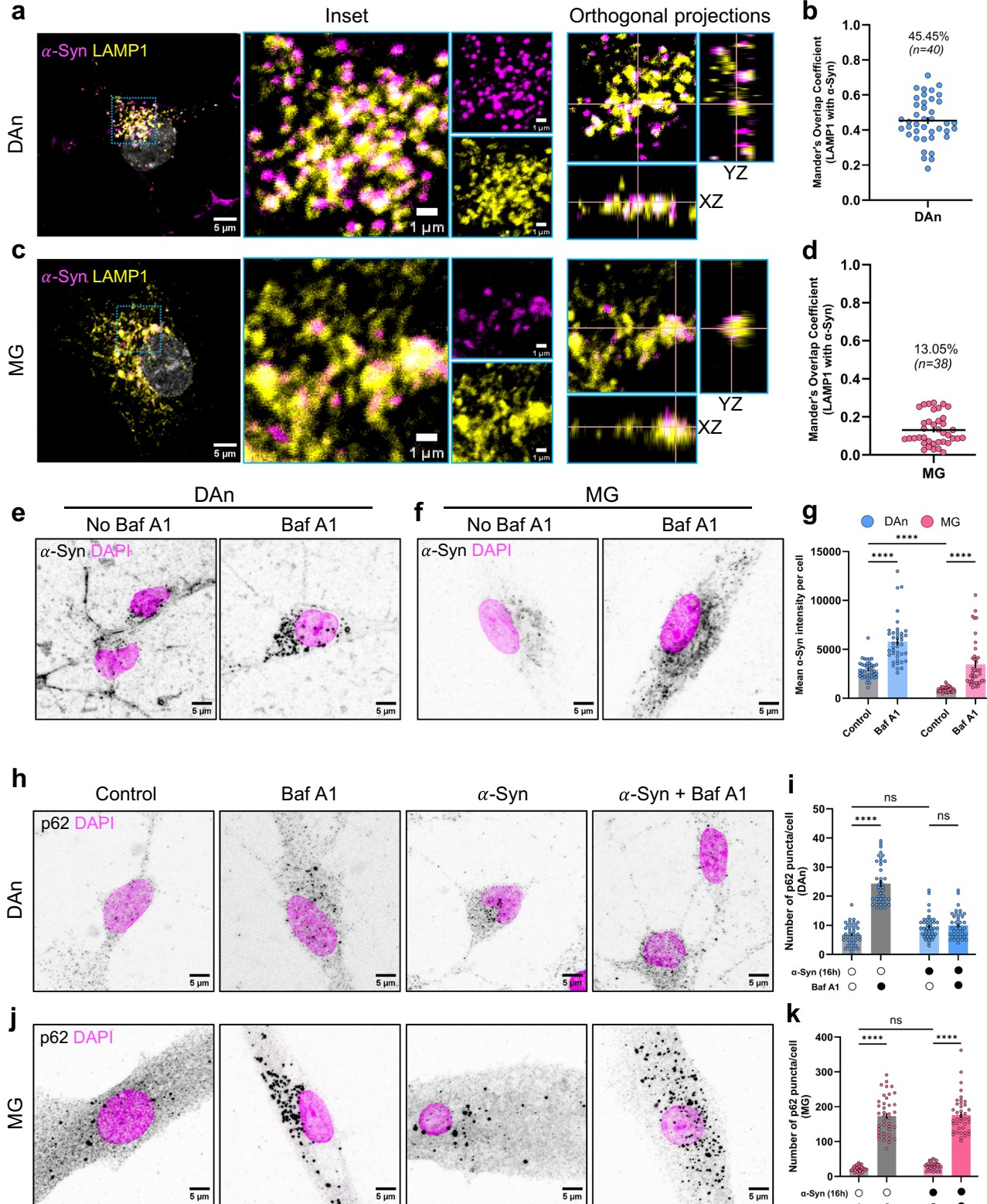

suggest that α-Syn−burdened neuronal cells influence bystander microglial autophagy flux, possibly through secreted factors.

While microglia employ different mechanisms, such as phagocytosis for clearance of α-Syn aggregates[60,61], autophagy is one of the pathways for clearing neuron-derived α-Syn[18]. To assess the relevance of upregulated autophagy in bystander microglia, we tracked the fate of aggregates post-transfer and observed 38.57% of transferred aggregates co-localized with p62 in microglial cells (Fig. 7g–i),

suggesting that neuronal cell-derived aggregates are destined for degradation in microglia. Supporting this, neuronal cells had reduced aggregate burden when co-cultured with microglia−both the number of α-Syn particles and the overall load of aggregates (number of α-Syn particles per unit area of a cell) were significantly reduced compared to mono-culture conditions (Fig. 7j, k). These findings highlight a critical bystander response of microglia in clearing neuronal cell-derived α-Syn aggregates.

**Fig. 5 | Validation of the neuronal and microglial responses to α-Syn using hiPSC-derived cells. a** Representative images of α-Syn colocalization with LAMP1+ lysosomes in control neurons (DAn). Blue dashed box depicts region in the inset, orthogonal projection of which is shown. **b** Quantification of the percentage of lysosomes overlapping with α-Syn in control neurons. $N = 3$ independent experiments, $n = 40$ cells. Mean percentage mentioned within the graph. **c** Representative images of α-Syn colocalization with LAMP1+ lysosomes in control microglia. Blue dashed box depicts region in the inset, orthogonal projection of which is shown. **d** Quantification of the percentage of lysosomes overlapping with α-Syn in control microglia. $N = 3$ independent experiments, $n = 38$ cells. Mean percentage mentioned within the graph. Representative images of α-Syn load in control neurons (**e**) and control microglia (**f**) in control and bafilomycin A1 treated conditions. **g** Quantification of the mean fluorescence intensity of α-Syn in different conditions. $N = 3$ independent experiments, $n = 40$ cells per group. Error bars represent SEM. Statistical significance was analyzed using Two-Way ANOVA with uncorrected

Fisher's LSD. DAn: Control versus Baf A1, ****$p = 5.79 \times 10^{-12}$; MG: Control versus Baf A1, ****$p = 5.43 \times 10^{-10}$; Control: DAn versus MG, ****$p = 5.96 \times 10^{-7}$. **h** Representative images of p62 immunostaining in control neurons in different conditions. **i** Quantification of the number of p62 puncta per cell in control neurons. $N = 3$ independent experiments, $n = 40$ cells per group. Error bars represent SEM. Statistical significance was analyzed using Two-Way ANOVA with Tukey's multiple comparison. Control versus Baf A1, ****$p < 1 \times 10^{-15}$; control versus α-Syn, ns ($p = 0.0876$); α-Syn versus α-Syn+Baf A1, ns ($p = 0.9773$). **j** Representative images of p62 immunostaining in control microglia in different conditions. **k** Quantification of the number of p62 puncta per cell in control microglia. $N = 3$ independent experiments, $n = 40$ cells per group, except for only α-Syn treated group, for which $n = 38$ cells. Error bars represent SEM. Statistical significance was analyzed using Two-Way ANOVA with Tukey's multiple comparison. Control versus Baf A1, ****$p < 1 \times 10^{-15}$; control versus α-Syn, ns ($p = 0.8507$); α-Syn versus α-Syn+Baf A1, ****$p < 1 \times 10^{-15}$. Source data are provided as a Source data file.

## Discussion

### Role of TNTs in aggregate spreading

Several studies have reported that amyloidogenic aggregates can induce the formation of intercellular connections such as tunneling nanotubes (TNTs), which facilitate their spread between cells. Initially, this was demonstrated with aggregated prion protein (PrP^Sc) between CAD neuronal cells and dendritic cells[62]. More recently, research has found that various aggregates, including Amyloid-β, Tau, α-Syn, and mutant Huntingtin (mHTT), increase intercellular connections, further promoting aggregate transfer[21]. Notably, DNTs, actin-rich connections between neuronal dendrites in the mammalian cortex, have been observed at increased density prior to amyloid plaque formation in APP/PS1 Alzheimer's disease mouse models, and can mediate intercellular transport of amyloid-β[23]. Moreover, TNT-like connections are increased in vitro by the small GTPase, Rhes[63]. Given its enrichment in the striatum, Rhes also facilitates mHTT spread in vivo from the striatum to cortical areas[64]. In vitro, α-Syn aggregates are transferred more efficiently from mouse neurons to astrocytes than in the opposite direction[28]. Such transfer has also been demonstrated between murine primary cortical neurons and microglia[55]. Similarly, human neuronal cells transfer α-Syn aggregates to microglia more effectively via TNTs than they receive them[30]. These studies suggest a conserved mechanism allowing neurons to offload their aggregate burden via TNTs.

In this study, we identify autophagy dysfunction in neurons as a key driver of TNT-mediated α-Syn transfer to microglia. This provides a mechanistic insight into the directionality of aggregate transfer, while revealing differential impacts of α-Syn aggregates on the lysosomal and autophagic pathways in neuronal versus microglial cells.

### Differential effect of α-Syn aggregates on lysosome function and autophagic pathways in neuronal cells versus microglia

Impaired autophagy is a well-established molecular hallmark of diseases[65]. Clearance of aggregate-prone proteins such as α-Syn relies predominantly on the autophagy-lysosome pathway, although the ubiquitin-proteasome system also plays a role[66,67]. Inhibition of autophagosome-lysosome fusion with bafilomycin A1 exacerbates α-Syn-associated cytotoxicity[68], an effect attributed to reduced levels of SNAP29—the SNARE protein required for autophagosome-lysosome fusion—in α-Syn-transduced cells[12]. In vivo, lysosomal dysfunction is an early feature of dopaminergic neuron loss in MPTP models of Parkinson's disease, accompanied by reduced lysosome number and autophagosome accumulation; these phenotypes can be rescued by rapamycin-driven autophagy activation[69]. Consistently, decreased lysosome-associated markers have been detected in brains from human PD subjects[70], and impaired lysosomal activity has been observed both in α-Syn-challenged neuronal cells and in PD patient-derived dopaminergic neurons[25,71]. Notably, lysosomes containing

α-Syn can themselves be transferred between cells via TNTs[25,26]. Together, these findings suggest that α-Syn aggregates impose a substantial burden on the autophagic machinery, which may foster their intercellular dissemination.

In line with this, our results show that α-Syn induces markedly greater lysosomal damage in neuronal cells than in microglia, reflected by increased aggregate association with neuronal lysosomes and an impaired degradative capacity that contrasts sharply with the preserved lysosomal function of microglia. This differential response was conserved in hiPSC-derived neurons and microglia treated with α-Syn PFFs, highlighting a robust and consistent biological distinction between the two cell types across experimental systems. These findings are in line with recent in vitro and in vivo data showing differential uptake and processing of α-syn fibrils: following injection of fluorescently labeled fibrils, glial cells initially accumulate the aggregates and clear them rapidly, whereas neurons show lysosomal impairment and retain fibrils more stably and at a slower rate[72]. The enhanced resilience of microglial lysosomes likely stems from their macrophage-like identity. Although microglial lysosomes are less acidic than those of peripheral macrophages[73,74], their acidity and degradative potential can increase in inflammatory contexts—for example, during fibrillar clearance[73] or in response to TLR2-mediated autophagy activation[75]. It is therefore plausible that α-Syn elicits a pro-inflammatory or stress-induced state that enhances microglial lysosomal function, although this will require further experimental validation.

Additional mechanisms may also contribute to the robust lysosomal response in microglia. BV2 microglia exposed to fibrillar α-Syn recruit TANK-binding kinase 1 (TBK1) and Optineurin (OPTN) to lysosomes, promoting clearance of damaged organelles through lysophagy[76]. Consistent with this, we observed elevated lysophagy and pronounced lysosome biogenesis in microglia compared to neuronal cells. Moreover, α-Syn exposure resulted in mTOR inhibition—potentially via galectin-8–mediated sensing of lysosomal damage[77]—which could facilitate TFEB nuclear translocation. Interestingly, although TFEB translocated to the nucleus in both cell types, the downstream transcriptional response was substantially muted in neuronal cells. This discrepancy may reflect differences in chromatin accessibility, selective co-operation with other transcription factors such as TFE3 or FOXO3 in microglia, or divergent balances of co-activators and co-repressors. These possibilities remain speculative but underscore a fundamental difference in the capacity of neuronal cells and microglia to activate lysosomal repair programs.

Taken together, these findings support the conclusion that microglia mount a more effective autophagic and lysosomal response to α-Syn aggregates than neuronal cells. We consistently observed upregulated autophagy in microglial cell lines and hiPSC-derived microglia, whereas neuronal cell lines and hiPSC-derived neurons

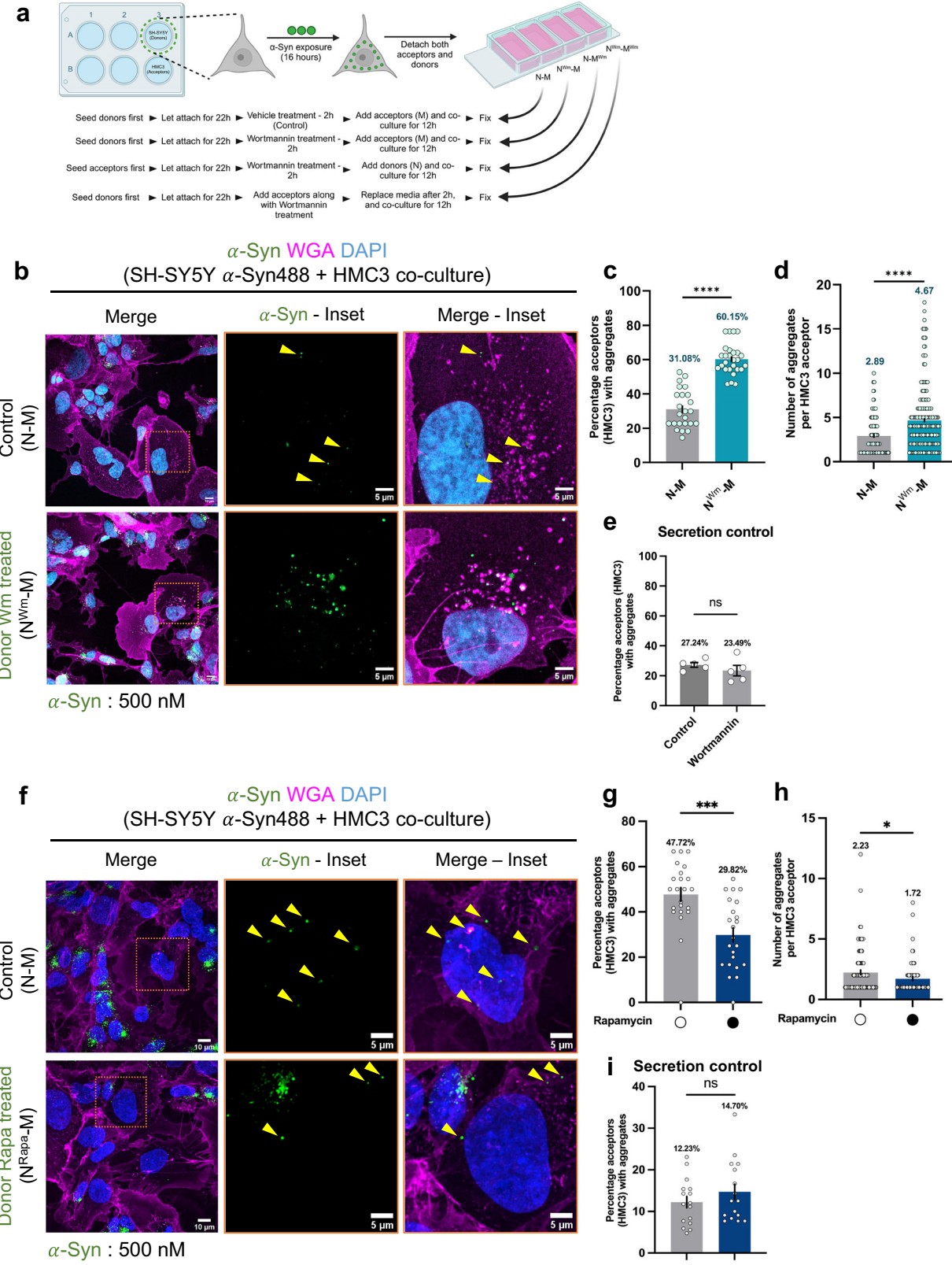

displayed defective autophagic flux and accumulation of substrates such as p62. Enhanced co-localization of aggregates with LC3, p62, and LAMP1 in microglia further corroborates earlier observations from BV2 cells exposed to fibrillar α-Syn[76]. These differences reveal that neuronal cells are substantially less capable of clearing α-Syn aggregates, creating a permissive environment for proteostatic failure.

This notion is supported by recent work showing that microglia efficiently degrade neuron-derived α-Syn aggregates in a p62-dependent manner[18]. In that study, activation of the Toll-like receptor 4 (TLR4)-NF-κB axis enhanced microglial autophagy and aggregate clearance. NF-κB, best known for its role in inflammatory cytokine production during α-Syn PFF exposure[78], also induces autophagy-

**Fig. 6 | Contribution of autophagy in transfer of aggregates. a** Schematic representation of co-culture strategy. Created in BioRender. Palese, F. (2026) https://BioRender.com/3zimbkm. **b** Representative images of co-cultures between $\alpha$-Syn loaded SH-SY5Y and HMC3 cells. Dotted box indicates the region of acceptor microglia zoomed in. **c** Percentage of HMC3 acceptor cells that received aggregates. $N = 3$, $n = 240$ acceptor cells for both groups. Error bars represent SEM. Statistical significance was analyzed using two-sided Mann-Whitney U test. ****$p = 4.51 \times 10^{-13}$. **d** Number of aggregates received per acceptor HMC3 in (c). $N = 3$, $n = 137$ acceptor cells for N-M, 196 acceptor cells for $N^{Wm}$-M. Error bars represent SEM. Statistical significance was analyzed using two-sided Mann–Whitney U test. ****$p = 2.94 \times 10^{-7}$. **e** Secretion control of aggregate transfer from neuronal cells (wortmannin treated or not) to microglia. Error bars represent SEM. $N = 3$, $n = 191$ cells for control and 259 cells for wortmannin-treated condition. Statistical significance was measured using two-sided unpaired Student's t-test with Welch's correction. ns: $p = 0.3664$. **f** Representative images of 12 h co-cultures between $\alpha$-Syn loaded SH-SY5Y and HMC3 cells. Rapamycin treatment represented by the superscript Rapa. **g** Percentage of HMC3 acceptor cells that received aggregates. $N = 3$, $n = 195$ acceptor cells for N-M, 200 acceptor cells for $N^{Rapa}$-M. Error bars represent SEM. Statistical significance was analyzed using two-sided Mann–Whitney U test. ***$p = 1.167 \times 10^{-4}$. **h** Number of aggregates received per acceptor HMC3 in (g). $N = 3$, $n = 94$ acceptor cells for N-M, 61 acceptor cells for $N^{Rapa}$-M. Error bars represent SEM. Statistical significance was analyzed using two-sided Mann–Whitney U test. *$p = 0.0425$. **i** Percentage of HMC3 acceptor cells that received aggregates in control and rapamycin-treated secretion controls. $N = 3$, $n = 272$ acceptor cells for N-M, 193 acceptor cells for $N^{Rapa}$-M. Error bars represent SEM. Statistical significance was analyzed using two-sided Mann–Whitney U test. ns: $p = 0.3549$. Source data are provided as a Source data file.

related gene expression[79,80]. It is therefore plausible that microglial autophagy upregulation in our system reflects a similar NF-$\kappa$B-dependent response, although this remains to be formally tested. Overall, these observations delineate a sharp contrast in how neuronal cells and microglia respond to $\alpha$-Syn aggregates: neurons experience lysosomal stress and autophagic dysfunction, whereas microglia activate pathways that maintain or enhance degradative capacity. This disparity provides a compelling rationale for why neurons may rely on intercellular transfer—particularly via TNTs—to relieve themselves of aggregate burden.

**Autophagy dysregulation drives aggregate transfer**

Given the significant impact of aggregate accumulation on autophagy in neuronal cells, we hypothesized that this impairment could contribute to the transfer of aggregates to microglial cells[30]. While we observe a significant reduction in the extent of $\alpha$-Syn transfer upon rapamycin treatment of aggregate-burdened neuronal cells, inhibition of autophagy by wortmannin resulted in a notable increase in aggregate load and transfer to microglia, underscoring the role of dysfunctional autophagy in the dissipation of aggregates to neighboring cells. Additionally, wortmannin treatment of acceptor microglia (N-$M^{Wm}$), or both donors and acceptors ($N^{Wm}$-$M^{Wm}$), resulted in an increased detection of $\alpha$-Syn-aggregates in microglia, likely reflecting the cumulative effect of elevated transfer and accumulation of aggregates not targeted for degradation. Assuming a linear relationship between autophagy inhibition and aggregate transfer, a predictive value for the average number of aggregates transferred (calculated by adding the effects of wortmannin on individual cell populations) was 10.94 ($N^{Wm}$-M + N-$M^{Wm}$ groups), closely matching the observed value of 11.84 ($N^{Wm}$-$M^{Wm}$ group). Nevertheless, it is yet to be determined whether there is bi-directional communication between neuronal cells and microglia that influences the extent of contact-mediated aggregate transfer upon autophagy inhibition. Notably, autophagy inhibition consistently enhances $\alpha$-synuclein transfer to microglia, irrespective of the neuronal aggregate load, suggesting a regulated off-loading mechanism rather than a passive effect of higher $\alpha$-synuclein burden. Of interest, a recent study highlights a non-cell autonomous mechanism in Huntington's disease (HD) and tauopathies, where activated microglia secrete proinflammatory chemokines (CCL-3/-4/-5) that suppress neuronal autophagy via mTORC1 activation through CCR5[59]. Our findings suggest that aggregate-laden neuronal cells with impaired autophagic flux induce a compensatory autophagic response in microglia. Microglia cultured with aggregate-loaded neuronal cells displayed enhanced autophagy. This phenotype was also observed when microglial cells were exposed to neuronal conditioned media. These results highlight a non-cell autonomous mechanism of aggregate clearance. It is also possible for neuronal cells to transfer $\alpha$-Syn-containing p62+ autophagosomes to microglia, which could not be ruled out by our experimental approach. The precise mechanism of non-cell autonomous autophagy response in microglial cells in the presence of $\alpha$-Syn-burdened neuronal cells remains to be determined.

Additionally, aggregate transfer from microglia to neurons, typically a less-efficient phenomenon, also increased upon autophagy inhibition, supporting the hypothesis that autophagy dysfunction serves as a signaling mechanism for intercellular aggregate transfer. Furthermore, autophagy inhibition using wortmannin and bafilomycin A1 significantly elevated intercellular connections between neuronal and microglial cells. Co-culture of wortmannin-treated neuronal cells and microglia also increased the extent of DiD-labeled vesicle transfer to microglia, highlighting the functionality of these additional connections. An increase in TNTs can be attributed to the accumulation of superfluous materials caused by autophagy inhibition, which triggers a global stress response promoting TNT formation[81]. While oxidative stress has been reported to upregulate TNTs, we did not observe a significant increase in intracellular ROS in neuronal and microglial cells upon autophagy inhibition, highlighting the roles of other players in the process, which remains to be elucidated.

Mechanistically, autophagy is closely linked to the endoplasmic reticulum (ER) stress response. While ER stress typically induces autophagy, this regulation is disrupted under several pathological conditions, including neurodegenerative diseases[82]. ER stress activation leads to cytosolic $Ca^{2+}$ release via the inositol trisphosphate (IP3) receptor. Elevated cytosolic $Ca^{2+}$ levels have been shown to positively regulate TNT formation. In particular, $Ca^{2+}$/calmodulin-dependent protein kinase II (CaMKII), a key transducer of the Wnt/$Ca^{2+}$ pathway, regulates TNT formation in neuronal cells through the actin-binding activity of its $\beta$-isoform[83]. Additionally, impairment of autophagosome–lysosome fusion has been reported to induce lysosomal membrane permeabilization (LMP)[84]—a phenomenon we observed to be significantly more prominent in neuronal cells than in microglia, leading to autophagosome accumulation. Interestingly, this accumulation was reversed by treatment with BAPTA-AM, a calcium chelator, suggesting a role for $Ca^{2+}$ in this process.

Although the precise mechanism of TNT formation remains incompletely understood, it is well established that actin polymerization into long, parallel filaments is a critical early step. These filaments allow TNTs to span distances beyond those of typical cellular protrusions, forming stable intercellular connections[56]. This actin-driven architecture facilitates the transfer of cargo such as $\alpha$-Syn aggregates and is also regulated in part by the Rac–PAK signaling pathway, which may itself be activated by aggregates[55]. In this context, we observed a reduction in the levels of the branched actin nucleator ARP3 and its upstream regulator CDC42 (although not statistically significant) in neuronal cells—but not in microglia—following autophagy inhibition. This finding supports the notion that the molecular pathways regulating TNT formation are cell-type specific. Notably, CDC42 has been reported to negatively regulate TNT formation in mouse neuronal cells[85], but to positively regulate it in macrophages[86]. Such divergent roles likely reflect intrinsic differences in cellular behavior: microglial

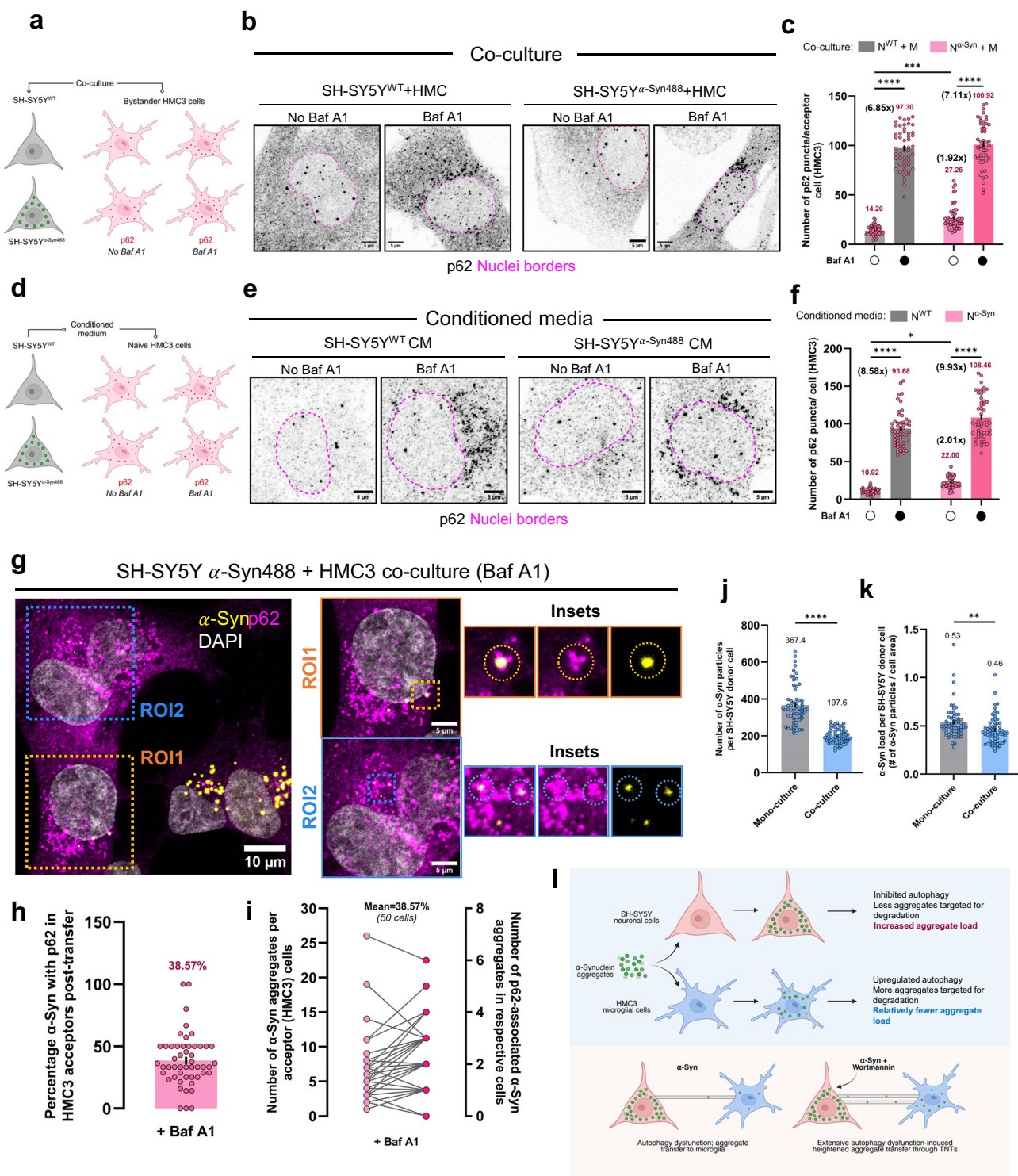

cells are more migratory and rely heavily on dynamic actin remodeling for morphological surveillance, phagocytosis, and other immune functions—processes maintained in part by the ARP2/3 complex[87,88]. Therefore, a strong reduction of ARP3 or CDC42 in microglia could be detrimental to these essential functions. In contrast, in neurons, reduced ARP3 and CDC42 levels may favor the formation of long, unbranched actin filaments, which support TNT elongation. It is also plausible that TNT formation in microglia is governed by distinct actin regulatory pathways, potentially requiring a finely tuned balance between actin polymerizing and depolymerizing factors, a hypothesis that warrants further investigation.

Our study reveals a mechanism underlying elevated aggregate transfer between cells, emphasizing the role of autophagy dysfunction in the propagation of aggregate-prone proteins. It also lays the foundation for further exploration of lysosome-regulated TNT-mediated communication between cells, in both physiological and pathological contexts. These findings underscore the importance of intercellular communication and autophagic flux modulation in the pathogenesis of neurodegenerative diseases, where impaired degradation of $\alpha$-Syn in neurons drives TNT-mediate aggregate transfer to microglia, which attempt to mitigate neuronal damage through enhanced autophagic activity. Targeting autophagy pathways may offer therapeutic

**Fig. 7 | Bystander autophagy effect and fate of $\alpha$-Syn aggregates post-transfer.**
**a** Schema depicts the co-culture paradigm, followed by immunostaining against
p62. Created in BioRender. Palese, F. (2026) https://BioRender.com/3zimbkm.
**b** Representative p62 immunofluorescence in HMC3 acceptor cells. **c** Number of
p62 puncta per microglial acceptor cell in co-culture conditions. N represents
neuronal cells and M microglia. $N = 3$ independent experiments, $n = 50$ cells per
group. Error bars represent SEM. Statistical significance was analyzed using Two-
Way ANOVA with Šídák's multiple comparison. ***$p = 0.0007$; $N^{WT} + M$: control
versus Baf A1, ****$p = 7.6 \times 10^{-14}$; $N\alpha^{-Syn} + M$: control versus Baf A1, ****$p = 7.6 \times 10^{-14}$.
**d** Schema depicts the experimental paradigm to assess secretory influence on
microglial autophagy, followed by immunostaining against p62. Created in BioR-
ender. Palese, F. (2026) https://BioRender.com/3zimbkm. **e** Representative p62
immunofluorescence in HMC3 cells. **f** Number of p62 puncta per microglial cell
exposed to different conditioned media. N represents neuronal cells and M
microglia. $N = 3$ independent experiments, $n = 50$ cells per group. Error bars
represent SEM. Statistical significance was analyzed using Two-Way ANOVA with
Šídák's multiple comparison. *$p = 0.0181$; $N^{WT}$ CM: control versus Baf A1,
****$p = 7.6 \times 10^{-14}$; $N\alpha^{-Syn}$ CM: control versus Baf A1, $p = 7.6 \times 10^{-14}$. **g** Representative
images of $\alpha$-Syn-loaded neuronal cells in co-culture with acceptor microglia
immunostained for p62. **h** Percentage of $\alpha$-Syn colocalized with p62 in microglia
post-transfer from neuronal cells. Error bar represents SEM. **i** Similar quantification
as in (**h**), but with details of the number of $\alpha$-Syn recognized by p62 per acceptor
cells (right y-axis) and the number of total aggregates received (left y-axis). For both
(**h**, **i**), $N = 3$, $n = 50$ acceptor cells analyzed. **j** Quantification of the number of $\alpha$-Syn
aggregates in donor SH-SY5Y cells when in monoculture, or co-culture with HMC3.
$N = 3$, $n = 60$ cells. Error bars represent SEM. Statistical significance was analyzed
using two-sided Mann–Whitney test. ****$p < 0.0001$. **k** Overall load of $\alpha$-Syn
aggregates per neuronal cell in monoculture or co-culture with microglia. $N = 3$,
$n = 60$ cells. Error bars represent SEM. Statistical significance was analyzed using
two-sided Mann–Whitney test. **$p = 0.0019$. **l** Schematic representation of the
major findings of this study. Created in BioRender. Palese, F. (2026) https://
BioRender.com/3zimbkm. Source data are provided as a Source data file.

potential to not only reduce the burden of $\alpha$-Syn aggregates at an intra-
cellular level but also limit the inter-cellular spread of toxic aggregates,
thereby mitigating the progression of the disease. While the precise
mechanisms linking autophagy inhibition to increased TNT formation
and transfer remain to be fully elucidated, our observations may
extend beyond neuron–microglia interactions to encompass
neuron–astrocyte communication, providing a more comprehensive
view of the cellular processes underlying these diseases.

**Limitations of the study.** Due to the inherent difficulty of transfecting
microglial cells for overexpression or knockdown studies, we con-
ducted all experiments without genetic perturbations, relying instead
on relevant and well-characterized pharmacological agents. While we
validated our findings in iPSC-derived neurons and microglia, future
studies employing advanced 3D cellular models (such as brain orga-
noids and assembloids) as well as complementary animal models
carrying patient-specific or autophagy-related gene mutations, will be
essential to confirm these results in more complex systems and to
further assess their relevance to human disease pathogenesis.

## Methods
### Culture of cell lines
Human neuroblastoma cell line SH-SY5Y (referred to as neuronal cells
in the manuscript, derived from SK-N-SH neuroblastoma line) were
cultured in RPMI1640 media (Euroclone, ECB2000L), and supple-
mented with 10% fetal calf serum (FCS) (Eurobio Scientific, CVFSVF00-
01) and 1% penicillin-streptomycin (Gibco, 15140-122; 100 units/mL
final concentration). The human microglia clone 3 cell line (HMC3)
were cultured in DMEM (Sigma-Aldrich, D6429) supplemented with
10% FCS and 1% penicillin-streptomycin. Cells were cultured in a
humidified $CO_2$ incubator at 37 °C, and passaged using 0.05% Trypsin-
EDTA (Gibco, 25300-054) when they reached 90% confluency. Cells
were counted before each experiment, and seeded on autoclaved,
12 mm glass coverslips (uncoated) (Epredia, CB00120RA120MNZ0) for
fixed-cell imaging, and on 35 mm glass-bottom microdishes (Ibidi
GmbH, 81156) for live-cell imaging. Cells between passages 3 and 10
were used for experiments.

### Cell culture experiments with hiPSC lines
All procedures adhered to Spanish and EU guidelines and regulations
for research involving the use of human pluripotent cell lines. The
human iPSC lines used in our studies were generated following pro-
cedures approved by the Commission on Guarantees concerning the
Donation and Use of Human Tissues and Cells of the Carlos III Health
Institute, Madrid, Spain.

For both neurons and microglia, control hiPSC lines were used
(SP11 and SP13wt/wt, respectively). The hiPSC line generation and

characterization is as previously described[89,90]. hiPSC were maintained
on Matrigel (Corning, 354234)-coated plastic plates (Thermo Fisher
Scientific) and in mTeSR™1 medium (Stem Cell Technology, 85850)
until the start of the protocols of differentiation.

The employed protocol for the differentiation of microglia and its
characterization are described in detail in ref. 52. Briefly, hiPSC were
passed in single colonies and, after 2–4 days, mTeSR™1 medium was
supplemented with 80 ng/mL of Bone Morphogenetic Protein (BMP)-4
(PeproTech, 120-05) for a total of 4 days. From the following day, cells
were changed using SP34 medium (StemPro™-34 SFM (Gibco™,
10639011), 1% of P/S (Cultek, SV30010), and 1% of Ultraglutamine (Glut;
Lonza, LZBE17-605EU1). For 2 days, media was supplemented with
80 ng/ml of VEGF (PeproTech, AF-100-20), 100 ng/ml of Stem Cell
Factor (SCF, PeproTech, 300-07), and 25 ng/ml of Fibroblast Growth
Factor (FGF)-2 (PeproTech, 100-18B). Then, from day 7 to day 14,
supplemented factors included 50 ng/ml of Fms-like tyrosine kinase
3-Ligand (Flt3-L, Humanzyme, HZ-1151), 50 ng/ml of IL-3 (PeproTech,
200-03), 50 ng/ml of SCF, 5 ng/ml of Trombopoietin (TPO, PeproTech,
300-18), and 50 ng/ml of M-CSF (PeproTech, 300-25). The last step
consisted on the addition to SP34 medium of Flt3-L, M-CSF, and 25 ng/
ml of GM-CSF (PeproTech, 300-03), changing the medium every
3–4 days. Starting from day 35, floating microglial progenitors were
collected from the culture's supernatant and passed through a 70 μm
Filcon™ Syringe-Type nylon mesh (BD Biosciences, 10271120). Cells
were counted and centrifuged at $300 \times g$ for 10 min. Recollected pro-
genitor microglial cells were plated at a final density of 5.000 microglia
in a well of a 24 well plate (Thermo Fisher Scientific) and on top of
plastic coverslips. Media was changed twice a week with Roswell Park
Memorial Institute (RPMI) 1640 Medium (Gibco™, 11875093) supple-
mented with 50 ng/mL of IL-34 (PeproTech,200-34) and M-CSF.
Microglia were considered mature after 1 week in culture.

NPCs were generated following a previously published protocol
from[91,92]. Briefly, iPSCs were split into a 96 well-plate, V-bottom shape,
and centrifuged $800 \times g$ for 10 min to force the aggregation. Cells were
grown on mTeSR medium (STEMCELL technologies, 05825) for 24 h.
Embryoid bodies (EBs) were plated in a 6 cm² dish and the medium was
then changed to Proneural [DMEM/F12 (Life, 21331-20) and Neurobasal
(Life, 21103-049)−1:1, 0.5% N2 (Life, 17502048), 1% B27 w/o Vitamin A
(Life, 12587-010), 1% L-Glutamine (Linus, X0551-100), 1% Penicillin/
Streptomycin (ScienCell, 0503), and 2-Mercaptoethanol (gibco 31350-
010)]. EBs were seeded in POLAM-coated (poly-L-ornitine Sigma-
aldrich, P4957; laminin, Sigma-Aldrich, L2020) wells of a 6 well plate
with Proneural supplemented with Noggin 200 ng/ml (PeproTech,
120-10 C) and SB431542 10 μM (TOCRIS, 301836-41-9). When NEP
rosettes were visible, they were enzymatically dissociated with trypsin
0.05% to obtain NPCs and plated on POLAM-coated wells of a 12 well
plate. NPCs were then split up to 6-8 times to purify the culture.

DAN neurons were differentiated following a previously published protocol[51]. NPCs at 80-100% confluency were cultured on POLAM-coated (poly-L-ornithine Sigma-Aldrich, P4957; laminin, Sigma-Aldrich, L2020) wells in DAn induction medium [DMEM/F12 (Life, 21331-20), 1% N2 (Life, 17502048), 1% Penicillin/Streptomycin (ScienCell, 0503)] supplemented with 200 ng/ml Sonic Hedgehog (PeproTech, 100-45) and 100 ng/ml FGF8 (PeproTech, 100-25) for 6 days. This step allowed for NPCs patterning towards dopaminergic fate. DAn progenitors were then plated on POLAM-coated dishes in N2B27 medium (DMEM/F12 (Life, 21331-20)−Neurobasal (Life, 21103-049) 1:1, 0.5% N2 (Life, 17502048), 1% B27(Life, 17504-044), 1% L-Glutamine (Linus, X0551-100), and 1% Penicillin/Streptomycin (ScienCell, 0503) for 10 days for maturation. For terminal differentiation, DAn were cultured on Matrigel (Corning, 354234) coated wells supplemented with 20 ng/ml BDNF (Peprotech, 450-02) and 20 ng/ml GDNF (Peprotech, 450-10) for 25 days.

## Preparation of α-Syn aggregates

α-Syn aggregates were prepared as described before[36]. Briefly, human wild-type α-Syn was purified from *Escherichia coli* BL21 (DE3) with RP-HPLC. Fibrils were then conjugated with Alexa Fluor™ 488 or 568 fluorophores (used in imaging experiments) (Invitrogen) using manufacturer's labeling kit or not tagged with a fluorophore. Fibrils were kept at −80 °C for long-term storage. Prior to exposure to cells, fibrils were diluted in growth medium at a working concentration of 500 nM and sonicated (BioBlock Scientific, Vibra Cell 7504) for 5 min at an amplitude of 80%, pulsed for 5 s "on" and 2 s "off". For iPSC-derived cells, owing to their fragile nature, a lower concentration of fibrils (200 nM) was used. Sonicated fibrils were then added on cells directly without the addition of any intracellular delivery agents for designated time points. Post-incubation, cells were washed with diluted trypsin solution (1:3, in 1× DPBS [Gibco, 14040-091]) and processed for experiments.

## Immunocytochemistry

Immunofluorescence on cells were performed using the standard protocol. Cells were fixed with 4% paraformaldehyde (PFA [Electron Microscopy Sciences, 15710]) for 30 min at room temperature (RT), followed by one wash with 1× DPBS. Cells were then incubated in 50 mM ammonium chloride (NH₄Cl [Sigma Aldrich, A0171]) solution for 15 min at RT to quench the fixative. Following three washes with 1× DPBS, cells were incubated in 1× DPBS-Tx (0.1% Triton-X100 [Sigma Aldrich, 9002-93-1] in 1× DPBS) for 5 min at RT. After three washes with 1× DPBS, cells were incubated in blocking solution of 2% bovine serum albumin (BSA [Sigma Aldrich, A9647]) prepared in 1× DPBS for 1 h at RT. Cells were then incubated with primary antibody overnight (for 16 h) at 4 °C. The following primary antibodies were used: mouse anti-LAMP1 (Developmental Studies Hybridoma Bank, H4A3; 1:100), rabbit anti-LC3 (Medical and Biological Laboratories International Corporation, PM036; 1:400), guinea pig anti-p62 (Progen, GP62-C; 1:400), and rabbit anti-TFEB (Cell Signaling Technology, 4240; 1:100), mouse anti-human Galectin3 (Clone 194804, MAB1154; 1:50), and rabbit anti-human IST1 (Proteintech, 19842-1-AP; 1:100). The next day, cells were washed three times with 0.1% DPBS-Tx, followed by incubation with respective secondary antibodies (all from Thermo fisher Scientific, dilution of 1:500 in blocking solution) for 1 h at RT. Cells were then washed three times with 1× DPBS, counterstained for nuclei with DAPI (1:1000 in PBS [Sigma-Aldrich, D9542]) and mounted on glass slides with Aqua-Poly/Mount (Polysciences Inc., 18606-20). Slides were imaged at least a day after mounting of coverslips.

hiPSC-derived cells were fixed with 4% PFA for 20 min, followed by three washes with 1× PBS. After three washes with 1× Tris-buffered saline (TBS) and 0.1% Triton-×100 (Sigma-Aldrich, 9036-19-5), cells were blocked with a solution composed of 0.3% Triton-×100 and 3% normal donkey serum (Millipore, 41105901) diluted in TBS for 2 h at RT. Cells were then incubated with primary antibodies in aforementioned dilutions overnight at 4 °C. The next day, cells were washed three times with 1× TBS and 0.1 % Triton-×100, and incubated with blocking solution for 1 h at RT. Secondary antibodies diluted in blocking solution were then added for 2 h at RT, followed by three washes with 1× TBS. Finally, cells were incubated with DAPI (Abcam, 228549), diluted to 1:5000 in TBS, and coverslips were mounted on glass slides with PVA-DABCO (Sigma Aldrich) mounting media.

## Lysosome degradative ability, pH, and motility assays

To measure the degradative ability of lysosomes, cells were grown on 35 mm glass-bottom microdishes for 24 h and first incubated with AlexaFluor647-conjugated Dextran of 10 kDa molecular weight (Invitrogen, D22914) for 3 h (pulse). After 3 h, cells were washed and replaced with media containing 500 nM α-Syn488, or only media for control conditions, and incubated for 16 h. DQ-BSA Red (Invitrogen, D12051) was diluted in fresh, complete medium to a final concentration of 10 µg/mL, and added to cells for the last 2 h of aggregate incubation at 37 °C. Cells were thoroughly washed with complete medium before imaging. Lysosomal pH was assessed using LysoSensor Green DND-189 (Thermo Fisher Scientific, L7535), diluted to 100 nM in fresh, complete medium and incubated for 30 min at 37 °C.

To assess the motility of lysosomes, cells were incubated for 16 h with α-Syn aggregates conjugated with AlexaFluor 568 fluorophore. After a thorough wash with 1:3 diluted trypsin, cells were incubated with 500 nM Lysotracker Green DND-26 (Thermo Fisher Scientific, L7526) diluted in fresh, complete medium for 30 min at 37 °C. After a wash with complete medium, fresh phenol red-free RPMI 1640 medium (for SH-SY5Y cells) and FluoroBrite™ DMEM (for HMC3 cells) was added and the dishes were incubated in a 37 °C, 5% CO₂, 50% relative humidity environmental chamber equipped with the microscope. Time-lapse imaging was performed using a Nikon Eclipse Ti2 microscope with a spinning disk confocal set-up (Yokogawa) with the following lasers (wavelength in nm): 405, 488, 561, and 640 nm. Cells were imaged using a 100× oil immersion objective (1.45 numerical aperture) for 5 min. 5 optical sections of 1 µm interval (±2 sections from the middle plane) were taken for each frame in two wavelengths−488 and 561 nm. Each acquisition was made at a time interval of 1.537 s.

## Autophagy and lysophagy flux analysis

To assess autophagy flux upon exposure of cells to α-Syn aggregates, cells were immunostained for p62 (as previously described). For bafilomycin treatment of α-Syn containing cells, 400 nM of the drug was added to cells in the 15 h (last 1 h of aggregate incubation), followed by fixation. For lysophagy analysis, similar approach was undertaken for aggregate and bafilomycin A1 incubation. Lysophagy was induced by the lysosome-damaging drug L-leucyl-L-leucine methyl ester (LLOMe; Sigma-Aldrich, L7393) at a concentration of 1 mM for 1 h, co-incubated with bafilomycin A1. Cells were then washed with 1× DPBS and processed for immunocytochemistry.

## Intracellular ROS quantification

Intracellular reactive oxygen species (ROS) levels were assessed following exposure to α-Syn aggregates and autophagic inhibition in SH-SH5Y and HMC3 cells. At the end of the standard α-Syn incubation protocol, in the presence or absence of wortmannin and bafiomycin A1 (see above), cells were incubated with CellRox™ Deep Red (5 µM; Invitrogen, C10422) for 30 min at 37 °C. as positive control for ROS induction, cells were treated with hydrogen peroxide (H₂O₂, 250 µM) for 1 h prior to CellROX incubation. Cells were subsequently washed three times with 1× DPBS, fixed with 4% PFA, and counterstained for nuclei with DAPI (1:1000 in PBS [Sigma-Aldrich, D9542]) and mounted on glass slides with Aqua-Poly/Mount (Polysciences Inc., 18606-20). Slides were imaged at least a day after mounting of coverslips.

## Immunoblotting

Both SH-SY5Y and HMC3 cells were grown in 6-well dishes for 24 h before incubation with untagged α-Syn aggregates (500 nM) for 16 h. 400 nM Bafilomycin A1 for the last 4 h before harvesting cells. Following incubation, cells were trypsinized, and the pellet fraction was collected. Cell pellets were homogenized in 100 µl of radio-immunoprecipitation assay (RIPA) buffer consisting of 50 mM Tris-HCl (pH 7.4), 1% TritonX-100, 0.5% sodium-deoxycholate, 0.1% sodium dodecyl sulfate, 150 mM sodium chloride, and 2 mM ethylenediaminetetraacetic acid, supplemented with 1× protease inhibitor (cOmplete Mini, EDTA-free; Sigma, 11836170001). Protein concentrations were measured using Bradford's method, following manufacturer's instructions (Thermo Fisher Scientific). Proteins (20 µg) were denatured in SDS (8%) and β-mercaptoethanol (5%) at 95 °C for 10 min. After separation by SDS-PAGE on a 14% gel, proteins were electrotransferred to nitrocellulose membranes. The membranes were blocked with 5% BSA and incubated overnight with primary antibodies (anti-LC3B, D11 XP #3868, Cell Signaling Technology, 1:1000; anti-phospho S211-TFEB, E9S8N #37681, Cell Signaling Technology, 1:1000; anti-phospho S235/236-S6 ribosomal protein, D57.2.2E XP #4858, Cell Signaling Technology, 1:1000; anti-S6 ribosomal protein, 5G10 #2217, Cell Signaling Technology, 1:1000; anti CDC42, ab64533, abcam, 1:1000; anti-ARP3, A5979, Sigma-Aldrich, 1:2000; anti-β-TUBULIN, PA1-41331, Thermo Fisher Scientific, 1:5000; and anti-GAPDH antibody Sigma, G9545, 1:5000) in blocking buffer, followed by incubation with horseradish peroxidase-conjugated secondary antibody (Millipore, 1:5000) in TBS containing 0.1% Tween-20 at room temperature for 1 h. Finally, proteins were visualized using an ECL kit (Thermo Fisher Scientific), and chemiluminescence images were acquired using GE Amersham Imager AI680 analyzer.

## mRNA isolation and qPCR

RNA was isolated using TRIZOL reagent (Invitrogen) following manufacturer's protocol. 1 µg of RNA was used to synthesize cDNA using high-capacity cDNA reverse transcription kit (4368814, Applied Biosystems). qPCR was carried out using diluted cDNA (20 ng per reaction) using iTaq universal SYBR Green supermix (1725124, Bio-Rad). Relative mRNA expression was calculated using the ΔΔCt method and each gene was normalized with Ct value of β-actin. Three technical replicate reaction mixes were performed for each gene and biological replicate. Primers (commercially available from OriGene Technologies) against the following human genes were used:

| Gene | Forward primer | Reverse primer |
| --- | --- | --- |
| LAMP1 | CGTGTCACGAA GGCGTTTTCAG | CTGTTCTCGTC CAGCAGACACT |
| CTSD | GCAAACTGCTG GACATCGCTTG | GCCATAGTGGATG TCAAACGAGG |
| CTSB | GCTTCGATGCACG GGAACAATG | CATTGGTGTGGAT GCAGATCCG |
| ATP6V1H | CGGGTCAATGA GTACCGCTTTG | GATACTGGAGCTG AAAGCCACAC |
| SQSTM1 | TGTGTAGCGTCT GCGAGGGAAA | AGTGTCCGTGT TTCACCTTCCG |

## α-Syn transfer assay

To assess for aggregate transfer, neuronal cells or microglia were grown for 24 h in a 6-well dish and then exposed to 500 nM of α-Syn aggregates, prepared as previously described. After 16 h of incubation, cells (now the donor population) were trypsinized and seeded on 12 mm coverslips. After 22 h post-seeding, donor cells were treated (or not) with 200 nM wortmannin (Sigma Aldrich, W1628) for 2 h. For rapamycin treatment, cells were treated with 100 nM of the drug for 6 h. Cells were then washed with complete media once, and then fresh media was added. At this point, microglia or neuronal cells (acceptor population) were added to the culture (1:1 ratio). For the experiment wherein both donor and acceptor cell populations were treated with wortmannin, the donor population was seeded first on coverslips and grown for 22 h. Media was replaced and the acceptor population was added, along with wortmannin. Media was replaced after 2 h of wortmannin treatment. Co-cultures were done for 12 h, after which cells were fixed to preserve TNTs (described below) and stained with wheat germ agglutinin 647 (WGA, Thermo Fisher Scientific, W32466; 3.33 µg/mL) for 15 min at RT, followed by nuclei counterstaining with DAPI (1:1000 in 1× DPBS).

To assess the extent of secretion-mediated aggregate transfer, conditioned media from aggregate containing cells treated (or not) with wortmannin was added to acceptor microglia for 12 h. Data represented is normalized to secretion control (data from secretion control subtracted from the co-culture data).

## TNT counting

To efficiently visualize TNTs in cultures, cells were fixed at sub-confluency (~70%). In order to appropriately preserve TNTs, two different fixative solutions were used, as described previously[30,93]: fixative 1 (0.05% glutaraldehyde [GA {Sigma Aldrich, G5882}], 2% PFA, 0.2 M HEPES buffer [Gibco, 15630-080] in 1× DPBS), followed by fixative 2 (4% PFA, 0.2 M HEPES buffer in 1× DPBS) for 15 min each at RT. Cells were then labeled with Phalloidin 647 (Thermo Fisher Scientific, A12380; 1:250 in 1× DPBS) and DAPI (1:1000 in 1× DPBS) for 15 min each at RT. Cells were then washed once with 1× DPBS and mounted on glass slides.

## Fixed-cell microscopy

Images were acquired using Zeiss LSM900 inverted confocal microscope equipped with four lasers (wavelength in nm): 405, 488, 561, and 640 nm. For TNT counting, samples were imaged using 40× oil immersion objective (1.3 numerical aperture) with a field-of-view effective zoom of 0.8×, whereas all other immunofluorescence samples were acquired using a 63× oil immersion objective (1.4 numerical aperture) with a 1× zoom. Image acquisition was performed using ZEN Blue software. The entire cell volume was imaged for all samples, with optical sections of 0.45 µm. Depending on the cell types, the entire volume of cells ranged between 7 and 13 µm in thickness.

Super-resolution images were acquired using Zeiss LSM 780 Elyra SIM set-up (Carl Zeiss, Germany) using Plan-Apochromat 63×/1.4 oil objective with a 1.518 refractive index oil (Carl Zeiss). 16-bit images were acquired in "frame-fast" mode between wavelengths, with 32.0 µm grid size. Optical thickness was set at 0.133 µm. Raw images were processed using the SIM processing tool of Zen black software.

## Quantification and statistical analyses

For TNT counting, images were processed for analysis using the manual TNT annotation plug-in of ICY software (https://icy.bioimageanalysis.org/plugin/manual-tnt-annotation/). Number of p62 puncta or α-Syn aggregates per cell were manually counted in 3D images, through the z-stacks. Colocalization analysis was performed using the JACoP plug-in FIJI[94]. 3D images were used for colocalizing two channels with object-based detection, and thresholding of objects was performed manually. For analyzing degradative abilities of lysosomes, selections were created from dextran channel, and intensity of DQ-BSA was measured in all lysosomes. To measure DQ-BSA intensity in lysosomes positive for α-Syn, image calculator function of FIJI was used, to identify pixels positive for both dextran and α-Syn (AND operation). The resultant image was thresholded and made to a selection, followed by masking on the DQ-BSA channel before measuring the intensity. Triple

colocalization events were counted manually for each cell in 3D images. Time-lapse images were analyzed for ~2 min (80 frames) to track lysosomal motility. Manual tracking was performed using the FIJI plug-in MTrackJ[95] for single lysosomes. Only those lysosomes were tracked that were not in the perinuclear area, and that remained within the region of interest for the entire duration of tracking without merging/interacting with other lysosomes. Movies were then created at a frame speed of 5 fps. Actin coherency was analyzed using OrientationJ plug-in of FIJI (developed by the Biomedical Imaging Group, EPFL, Switzerland).

No prior power analysis was done to measure the sample size. No data points were excluded from analyses. Graphs were plotted using GraphPad Prism 10.0, and appropriate statistical tests were performed on raw data. For all datasets, an initial normality distribution test was performed, and non-parametric tests were performed for any datasets that did not satisfy normal distribution. Statistical tests performed are mentioned in the figure legends, along with $p$-values. In cases of extreme statistical significance, GraphPad Prism displays $p$-values up to 15 decimal points, in which cases the lowest displayable values have been mentioned ($p < 1 \times 10^{-15}$). All the experiments were performed for three independent biological replicates. Investigators were not blinded to the groups during analysis.

### Reporting summary

Further information on research design is available in the Nature Portfolio Reporting Summary linked to this article.

## Data availability

All data are available in the main text, or in the supplementary materials. Source data are provided with this paper. Source data are provided with this paper.

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

## Acknowledgements

We thank Dr. Stephanie Maya and Dr. Valeria Valente for technical assistance with experiments. We thank Dr. Christel Brou, and all the members of the Membrane Traffic and Pathogenesis Unit, Institut Pasteur, for insightful discussions, Dr. Sevan Belian for critical reading of the manuscript. We thank Dr. Gautham Hari Narayana Sankara Narayana for technical assistance with actin orientation analysis. We thank Reine Bouyssie, a member of the administrative staff of the Membrane Traffic and Pathogenesis Unit for her continued support. R.C. was supported by Pasteur-Paris University (PPU) International Doctoral Program and The Journal of Cell Science Traveling Fellowship (JCSTF24101615). F.P. was supported by Fondation Alzheimer Jeunes Chercheurs Fellowship (990987) and the MSCA-IF-Fellowship (LipiSyn - 101064077). P.S. was supported by the Biology Summer Studentship Programme—Institut Pasteur and Trinity College, University of Cambridge. V.T. was the recipient of a pre-doctoral La Caixa INPhINIT Incoming Fellowship (code: LCF/BQ/DI21/11860038. J.M.M. was the recipient of a pre-doctoral fellowship FPI (PRE2022-104573) from the Spanish Ministry of Economy and Competitiveness (MINECO). A.C. acknowledges PID2022-139546OB-I00 supported by MCIN/AEI/10.13039/501100011033 and FEDER, and PDC2021-121051-I00 supported by MCIN/AEI/10.13039/501100011033 and by the European Union Next Generation EU/ PRTR). C.Z. is supported by France Parkinson—Soutien de l'Association France Parkinson 2021, Don Explore AD—Programme Explore de l'Institut Pasteur, Agence Nationale de la Recherche (ANR-20-CE13-0032-01 and ANR UnProSec ANR-23-CE16-0012-03), Fondation pour la Recherche Médicale (FRM EQU202103012692 and FRM MND202310017892), and Fondation Alzheimer (AAP 2024). Research was conducted within the context of Pasteur International Joint Research Unit Neurodegenerative Diseases.

## Author contributions

Conceptualization: C.Z. and R.C.; Methodology: R.C., F.P., P.S., V.T., J.M.M., S.S., T.N., and M.H.; Investigation: R.C., F.P., P.S., V.T., and J.M.M.; Visualization: R.C., F.P., V.T., and J.M.M.; Supervision: C.Z. and A.C.; Funding acquisition: C.Z. and A.C.; Writing—original draft: R.C.; Writing—review and editing: R.C., F.P., P.S., V.T., J.M.M., A.C., and C.Z.

## Competing interests

The authors declare no competing interests.
