## [Transparent Peer Review file · Nature Communications]

Impaired α -Synuclein aggregate clearance in neuronal cells drive their spread to microglia through tunneling nanotubes

Corresponding Author: Professor Chiara Zurzolo

Version 0:

Reviewer comments:

Reviewer #1

(Remarks to the Author)

The paper titled "Neuronal Autophagy Failure Drives α -Synuclein Transfer to Microglia for Aggregate Clearance" demonstrates a correlation between autophagy dysfunction in neuronal cells and TNT-mediated α -Synuclein transfer to microglia.

The study is mainly a correlation study. However, the authors claimed "Causative role of autophagy dysfunction in driving TNT-mediated α -Syn aggregate transfer from neuronal cells to microglia." What authors claim is a fundamentally very weak statement based on the experiments that have been shown. There is no mechanistic correlation via signalling pathway or at the molecular level between autophagy formation and the TNT formation pathway, which could provide the link for causation between autophagy and TNTs.

The study of TNT-mediated cell-to-cell transfer has been shown solely in cell lines, where the SHSY5Y cell is in an undifferentiated neuroblastoma condition, which possesses carcinogenic properties more than neurogenic properties. TNT-formation or TNTs between iPSC-derived neurons or microglia is not shown.

It is known that aggregation-clearing mechanisms are better and robust in microglia in comparison to terminally differentiated neurons. Microglia utilize multiple mechanisms for degradation, not solely relying on autophagy. Phagocytosis and other inflammation-linked pathways could also facilitate the clearance of neurodegenerative aggregates in microglia.

Previously, a few studies showed that a higher amount of aggregation accumulation caused more transfer to its neighbouring cells, even though none of the studies showed a strong reason for the transfer mechanism (exosomes, TNTs, or any other means).

Here are technical comments-

1) Line number 330 to 337, author explained:

"Interestingly, a comparable increase in aggregate transfer was observed in both experimental conditions of wortmannin treatment (Fig. S4C). This could be due to either increased transfer, and/or decreased clearance of aggregates in the treated microglia. To distinguish between these possibilities, we measured the aggregate load in microglia either immediately after 2h of wortmannin treatment, or after 12h of culture in drug-free media. We found an elevated load of aggregates in microglia following autophagy inhibition (Fig. S3A, right panels and C), consistent with the essential role of autophagy in the clearance of α -Syn aggregates."

The statement should be carefully clarified. In figure S3, fold change is compared, not the absolute aggregate loads between Neuronal and Microglial cells. Cell-to-cell transfer will depend on absolute aggregate load; autophagy inhibition by Wortmannin may increase alpha-synuclein load in microglia. However, the total load is visibly less as per the image Figure S3A. Microglia's degradation is not solely dependent on autophagy. Thus, microglia after 16h degrade a higher percentage of aggregates is logical.

2) Line 371: "autophagy impairment plays a functional role in regulating intercellular connections major route for α -Syn transfer between neuronal and microglial cells." is a fundamentally very weak statement. There is no mechanistic connection shown between the autophagy pathway and the TNT formation pathway.

3) Lines 395 to 397: "Taken together, these data suggest the influence of α -Syn-burdened neuronal cells on autophagy flux

of microglial cells via both contact-dependent and independent mechanisms.” Authors themselves have shown that this aggregation accumulation could be contact-independent. Thus, the whole paper that highlighted the correlation of TNT with autophagy is an indirect correlation.

- 4) Dextran is a fluid phase endocytosis marker; does it mediate alpha-synuclein internalization in neuronal cells and microglia through fluid phase endocytosis? Please give proper reference, otherwise show with other endocytosis inhibitors that it is not by other endocytosis mechanisms, but only through fluid phase endocytosis.
- 5) In iPSc cell line Figure 5, p62 and Lamp1 overlap in the absence/presence of Bafilomycin, not shown. Only p62 may not solely reflect lysophagy quantification.
- 6) Neuronal differentiation and microglia differentiation with markers are not shown in the paper, please provide the characterization of iPSC differentiation with markers.
- 7) TNTs were not labelled in the iPSC-derived neurons and microglia. It is important to know if IPSC differentiated cells also form TNTs similar to undifferentiated cell lines.
- 8) Line number 324, please check the figure number, would it be 6A instead of 5A?

Reviewer #2

(Remarks to the Author)

The manuscript by Chakraborty, Zurzolo and colleagues is very interesting, novel and generates new ideas in the field of autophagy and neurodegeneration. This study provides critical insights about how neurodegeneration-associated protein, which are transferred between cells via tunnelling nanotubes, could differentially affect autophagy in a cell-type specific manner. The authors elegantly demonstrate that microglial cells are more efficient in handling α -synuclein load and clearing them through autophagy, than the neuronal cells. Interestingly, they show that impairment of autophagy in neuronal cells causes transfer of α -synuclein to microglial cells for autophagic degradation. The study is undertaken by outstanding microscopy and relevant experimental systems to support the data. Overall, it is a beautiful story, and the manuscript is well written. The minor comments below will help to improve the manuscript that is already in good shape.

[1] Although lysosomal functionality is shown in Fig. 2, it would be nice to show additional data on lysosomal pH using LysoSensor and LysoTracker in neuronal (SH-SY5Y) and microglial (HMC3) cells.

[2] Although greater nuclear localization of TFEB is shown after α -synuclein treatment in both neuronal and microglial cells (albeit at different levels) in Fig. 3, increase in TFEB target gene expression only occurs in microglial cells. Explain why this is not seen in neuronal cells. Would be good to check mTOR activation (using S6 or S6 kinase phosphorylation) and TFEB phosphorylation by immunoblotting in both these cell-types in the presence or absence of α -synuclein.

[3] Fig. 4A: The LC3 immunoblotting data in bafilomycin clamp assay show inhibition of autophagosome formation in neuronal cells after α -synuclein treatment – which is correctly interpreted. However, the data in microglial cells might suggest an increase in autophagosome formation after α -synuclein treatment – although the authors state no change. Ideally, the quantification of LC3-II vs GAPDH in the Baf-treated conditions should be done in low exposure blots to capture any significant changes in LC3-II levels (to avoid quantifying overexposed bands due to autophagosome accumulation by Baf); whilst higher exposure blots can be used to quantify changes in no-Baf condition. Therefore, the LC3 blots should be re-quantified as suggested.

[4] Related to the p62 data in Fig. 4, the authors state that autophagic flux is upregulated in microglial cells after α -synuclein treatment. Based on the data presented, it is ideal to state that autophagic flux is not overtly perturbed in these cells. However, the original conclusion could remain (with some modification – like microglia are trying to upregulate autophagic flux for clearance) if the authors find increased autophagosome formation in Fig. 4A after re-quantification.

[5] The authors state that neuronal cells affect autophagy in microglial cells via α -synuclein transfer. For this, re-quantification of Fig. 4A is important. Additionally, in Fig. 6F and 6K, the authors could show the fold change of p62 in plus or minus Baf conditions in wild-type and α -synuclein conditions – current data is indicative of less p62 fold change in α -synuclein condition, which fits with the hypothesis. Here, can you show the levels of α -synuclein after co-culture to further support the notion about clearance of neuronal α -synuclein in microglia?

[6] A summary cartoon of the key findings arising from this study would be ideal in Fig. 7.

[7] Add information about LLOMe.

[8] Formatting: Word spacing is missing in few instances.

Version 1:

Reviewer comments:

Reviewer #1

(Remarks to the Author)

Reviewer's Report on the Revised Manuscript

I would like to thank the authors for their earnest effort for trying to improve the manuscript's quality. I began reviewing the revised manuscript and the rebuttal with high expectations, as it was from a top-tier TNT lab. However, I regret to say that the revised manuscript left me disheartened.

The authors tried to address the mechanistic link between autophagy impairment and TNT formation in neuronal and microglial cell lines. The authors added Supplementary Figure 8, where they have shown immunoblots of ARP3 and CDC42 in neuronal cells and microglia upon treatment with the autophagy inhibitor wortmannin (Wm) and flux blocker Baf A1. The authors claimed that in the figure Supplementary Figure 8a of WB, ARP3 and CDC42 in neuronal cells inhibited almost 50% (quantification in the Supplementary Figure 8b) upon autophagy inhibition by wortmannin (Wm) and flux blocker Baf A1. Loading control GAPDH of SH-SY5Y cells in the Wm-treated sample is visibly lower, and the normalized ARP3 and CDC42 levels cannot be around 50% as shown in the graph (Supplementary Figure 8b). In the represented WB ARP3 and CDC42 levels in Baf A1-treated sample show no reduction in band intensity compared to the control. None of the points in the quantitative graph, Supplementary Figure 8b, represents the WB figure of Supplementary Figure 8a. The mechanistic figure is the most important result to establish the causative role of autophagy inhibition in neuronal cells in the formation of TNTs towards microglia. However, authors represented the figure in the supplementary material, and the WB is not represented like other WBs of the manuscript with a Mw ladder, like Supplementary Figure 3a and d.

It is also not clear how Wortmannin (Wm) and flux blocker Baf A1, both can similarly affect ARP3 and CDC42 levels, where both Wm and Baf A1 pathways work with completely different molecular players.

Authors have shown a change in actin fibres directional orientation with autophagy inhibition, both in SHSY5Y cells and microglia cells; however, both cells show similar effects. There are no differences between the two cell types.

Authors further disagreed in the rebuttal point 3, they have given the following points to argue that correlation of TNT with autophagy is not an indirect correlation. The authors claimed direct correlation, stating that neuronal autophagy failure triggers the directional transfer of α -syn aggregates to microglia via TNTs as an alternative clearance route. However, the stated logics are weak. Please see the argument below.

- Autophagy inhibition in neurons (using wortmannin and bafilomycin A1) significantly increases TNT formation.

Autophagy inhibition increases aggregation/aberrant organelle load in neurons, which is known to increase oxidative stress and ROS. Several studies have shown that oxidative stress and ROS play a crucial role in TNT formation. Thus, the above data do not prove a direct correlation between autophagy and TNTs.

- At the molecular level, reduced levels of ARP3 and CDC42 and actin cytoskeletal reorganization (as shown in response to previous comments) are consistent with a cellular state favoring TNT formation.

The reviewer is sorry to state that the WB data are not solid, and this is explained above.

- Our experimental data indicate that secreted α -Syn contributes minimally to microglial aggregate load under these conditions, whereas direct transfer via TNTs is the dominant mechanism.

Increased accumulation of aggregates induces TNTs and TNT-mediated cell-to-cell transfer in many cell types. Several studies have shown that, and the lead author of this manuscript has earlier written a wonderful review on that in Victoria et al JCB 2017. This more reflects a direct correlation between the accumulation of aggregates with TNT-mediated transfer, not the autophagy inhibition with TNT formation.

TNTs between iPSC-derived neurons and microglia are not shown properly. The authors stated the following in rebuttal point 8.

"Given these limitations, quantifying TNTs in iPSC-derived cells in a robust and reproducible manner requires further protocol optimization. We are currently working on refining our imaging and analysis pipelines to enable reliable detection and quantification of TNTs in these more complex cellular environments."

Thank you for addressing the other questions. The reviewer is not convinced that neuronal autophagy failure is the cause that drives TNT-mediated α -Synuclein transfer to microglia to outsource aggregate clearance. This is a nice correlation study that shows accumulation of aggregates in the neuron drives TNT-mediated α -Synuclein transfer to microglia to outsource aggregate clearance. The authors established the correlation of aggregate accumulation with TNT beautifully. But the study is definitely not a study where one can say autophagy inhibition is the cause of TNT formation in neurons, and autophagy inhibition differentially induces TNTs in neurons and microglia.

Reviewer #2

(Remarks to the Author)

The authors have done nice revision work to address all the comments satisfactorily. The manuscript has is now improved and acceptable for publication.

Version 2:

Reviewer comments:

Reviewer #1

(Remarks to the Author)

Thank you to the authors for thoroughly addressing all the points. Now, the overall writing and the connection with autophagy inhibition are much clearer. The results now showed that "Additionally, aggregate transfer from microglia to neurons, typically a less-efficient phenomenon, also increased upon autophagy inhibition, supporting the hypothesis that autophagy dysfunction serves as a signaling mechanism for intercellular aggregate transfer." This is not a neuron-specific effect, although the impact on neurons is greater. The author also changed the title to better reflect these results.

We thank both the reviewers for their comments and suggestions towards our manuscript. We have replied below to all of them, performed new experiments, incorporated new data in the revised manuscript, and made several changes in the text, as outlined below.

Reviewer #1 (Remarks to the Author):

The paper titled “Neuronal Autophagy Failure Drives α -Synuclein Transfer to Microglia for Aggregate Clearance” demonstrates a correlation between autophagy dysfunction in neuronal cells and TNT-mediated α -Synuclein transfer to microglia. The study is mainly a correlation study. However, the authors claimed “Causative role of autophagy dysfunction in driving TNT-mediated α -Syn aggregate transfer from neuronal cells to microglia.” What authors claim is a fundamentally very weak statement based on the experiments that have been shown. There is no mechanistic correlation via signalling pathway or at the molecular level between autophagy formation and the TNT formation pathway, which could provide the link for causation between autophagy and TNTs.

The study of TNT-mediated cell-to-cell transfer has been shown solely in cell lines, where the SHSY5Y cell is in an undifferentiated neuroblastoma condition, which possesses carcinogenic properties more than neurogenic properties. TNT-formation or TNTs between iPSC-derived neurons or microglia is not shown.

It is known that aggregation-clearing mechanisms are better and robust in microglia in comparison to terminally differentiated neurons. Microglia utilize multiple mechanisms for degradation, not solely relying on autophagy. Phagocytosis and other inflammation-linked pathways could also facilitate the clearance of neurodegenerative aggregates in microglia.

Previously, a few studies showed that a higher amount of aggregation accumulation caused more transfer to its neighbouring cells, even though none of the studies showed a strong reason for the transfer mechanism (exosomes, TNTs, or any other means).

We appreciate the comments of this reviewer that have helped us to improve the quality and content of our manuscript. We have now added new experiments, and updated the graphs and performed analysis as suggested, which are detailed in the point to point reply below:

Here are technical comments-

1) Line number 330 to 337, author explained:

“Interestingly, a comparable increase in aggregate transfer was observed in both experimental conditions of wortmannin treatment (Fig. S4C). This could be due to either increased transfer, and/or decreased clearance of aggregates in the treated

microglia. To distinguish between these possibilities, we measured the aggregate load in microglia either immediately after 2h of wortmannin treatment, or after 12h of culture in drug-free media. We found an elevated load of aggregates in microglia following autophagy inhibition (Fig. S3A, right panels and C), consistent with the essential role of autophagy in the clearance of α -Syn aggregates.”

The statement should be carefully clarified. In figure S3, fold change is compared, not the absolute aggregate loads between Neuronal and Microglial cells. Cell-to-cell transfer will depend on absolute aggregate load; autophagy inhibition by Wortmannin may increase alpha-synuclein load in microglia. However, the total load is visibly less as per the image Figure S3A. Microglia's degradation is not solely dependent on autophagy. Thus, microglia after 16h degrade a higher percentage of aggregates is logical.

We thank the reviewer for this insightful and important comment. We agree that our initial representation of aggregate burden as fold change may have limited interpretability in the context of cell-to-cell transfer. In response, we have revised the relevant graphs (now in Fig. S4) to show absolute mean intensity values of α -synuclein aggregates in neuronal cells (Fig. S4c) and microglia (Fig. S4d), instead of fold change.

Additionally, we now include quantification of the total α -synuclein load across the two cell types, demonstrating that neuronal cells carry a significantly higher aggregate burden compared to microglia (new Fig. S4b). These data support our hypothesis that neuronal cells, which exhibit less efficient clearance capacity, may rely on directional aggregate transfer to microglia as an outsourcing mechanism for aggregate removal. This is now added in lines 348-351.

We fully agree with the reviewer's point that microglia use multiple degradative pathways beyond autophagy, and this is now clearly acknowledged in the revised text (lines 462-464). We have also clarified the interpretation of the effect of wortmannin in this context.

2) Line 371: “autophagy impairment plays a functional role in regulating intercellular connections major route for α -Syn transfer between neuronal and microglial cells.” is a fundamentally very weak statement. There is no mechanistic connection shown between the autophagy pathway and the TNT formation pathway.

We thank the reviewer for this important comment. To better address the mechanistic link between autophagy impairment and TNT formation, we have now performed western blot analyses for ARP3, a component of the ARP2/3 complex involved in branched actin nucleation, and CDC42, one of its upstream regulators.

Previous studies have reported that inhibition of ARP2/3 promotes the formation of long actin filaments, favoring TNT-like protrusions ¹, and that CDC42 signaling negatively regulates TNT formation between neuronal cells ², while having the opposite effect in other cell types such as macrophages ³. Consistently, our new data show that treatment with wortmannin or bafilomycin A1 led to a reduction in ARP3 and CDC42 levels in neuronal cells, but not in microglia. This correlates with an increase in TNT formation observed under autophagy-inhibited conditions.

These results suggest that autophagy inhibition may promote TNT formation in neuronal cells via modulation of actin regulatory pathways, particularly ARP2/3 and CDC42. We have incorporated this new finding into the revised Discussion section and provided relevant references.

Following is the excerpt from discussion section:

“In this context, we observed a reduction in the levels of the branched actin nucleator ARP3 and its upstream regulator CDC42 in neuronal cells—but not in microglia—following autophagy inhibition. This finding supports the notion that the molecular pathways regulating TNT formation are cell-type specific. Notably, CDC42 has been reported to negatively regulate TNT formation in mouse neuronal cells ², but to positively regulate it in macrophages ³. Such divergent roles likely reflect intrinsic differences in cellular behavior: microglial cells are more migratory and rely heavily on dynamic actin remodeling for morphological surveillance, phagocytosis, and other immune functions—processes maintained in part by the ARP2/3 complex ^{4,5}. Therefore, a strong reduction of ARP3 or CDC42 in microglia could be detrimental to these essential functions. In contrast, in neurons, reduced ARP3 and CDC42 levels may favor the formation of long, unbranched actin filaments, which support TNT elongation. It is also plausible that TNT formation in microglia is governed by distinct actin regulatory pathways, potentially requiring a finely tuned balance between actin polymerizing and depolymerizing factors, a hypothesis that warrants further investigation.”

The formation of TNTs requires the extension of long actin filaments that arrange themselves in a directional manner, implying that global reorganization of the actin cytoskeleton is necessary in cells that form more intercellular connections. To explore whether autophagy inhibition induces such reorganization, we performed orientation analysis of actin filaments ⁶ in neuronal and microglial cells upon wortmannin and bafilomycin A1 treatments. Our analysis revealed a pronounced directional arrangement of actin fibers in both cell types upon autophagy inhibition, indicating that autophagy compromise is associated to global rearrangement of actin cytoskeleton.

These findings are in accordance with a recent study that demonstrated that hyperosmotic stress-induced autophagy is dependent on the reorganization of actin cytoskeleton ⁷. Given the complex repertoire of actin regulatory proteins and the scope of this study, we did not further dissect the signaling pathways involved in TNT formation under conditions of autophagy impairment or lysosomal damage. These mechanisms are currently being investigated in an ongoing project in the lab.

These new results have been incorporated into Figure S8, and the Results (lines 416–435) and Discussion (lines 648–662) sections of the manuscript have been updated accordingly.

3) Lines 395 to 397: “Taken together, these data suggest the influence of α -Syn-burdened neuronal cells on autophagy flux of microglial cells via both contact-dependent and independent mechanisms.” Authors themselves have shown that this aggregation accumulation could be contact-independent. Thus, the whole paper that highlighted the correlation of TNT with autophagy is an indirect correlation.

We thank the reviewer for this comment. We agree with the inference that both contact-mediated and secretory factors are involved in the bystander autophagy response we observe in microglial cells. The aspect of secretory factors influencing autophagy in a non-cell autonomous manner has been discussed in the original submission. This part of the discussion has now been rephrased (lines 613-615).

We respectfully disagree with their statement “the whole paper that highlighted the correlation of TNT with autophagy is an indirect correlation”. A central conclusion of our study is that neuronal autophagy failure triggers the directional transfer of α -Syn aggregates to microglia via TNTs as an alternative clearance route. We provide multiple lines of evidence supporting this mechanism:

- Autophagy inhibition in neurons (using wortmannin and bafilomycin A1) significantly increases TNT formation.
- At the molecular level, reduced levels of ARP3 and CDC42 and actin cytoskeletal reorganization (as shown in response to previous comments) are consistent with a cellular state favoring TNT formation.
- Our experimental data indicate that secreted α -Syn contributes minimally to microglial aggregate load under these conditions, whereas direct transfer via TNTs is the dominant mechanism.

Taken together, we believe that our findings provide strong and convergent support for a functional link between impaired neuronal autophagy and contact-dependent, TNT-mediated aggregate transfer to microglia.

4) Dextran is a fluid phase endocytosis marker; does it mediate alpha-synuclein internalization in neuronal cells and microglia through fluid phase endocytosis? Please give proper reference, otherwise show with other endocytosis inhibitors that it is not by other endocytosis mechanisms, but only through fluid phase endocytosis.

We thank the reviewer for this comment. We are sorry for the confusion, but the rationale behind using dextran in the concerned experiment was to label lysosomes and not elucidate whether dextran would mediate the internalization of α -Syn aggregates. As described in the schematic representation of the experiment (Fig. 2a), α -Syn aggregates were added to the cell culture after removal of extracellular dextran in the media. This approach allowed for labeling lysosomes in a live-cell

experimental approach to further be able to assess for degradative ability of these organelles using DQ-BSA. To clarify this issue, we have now incorporated a few sentences and cited the studies that focused on understanding the mechanism of α -Syn internalization (lines 112-116). However, this is out of the scope of our study.

5) In iPSc cell line Figure 5, p62 and Lamp1 overlap in the absence/presence of Bafilomycin, not shown. Only p62 may not solely reflect lysophagy quantification.

We thank the reviewer for this comment, and agree that p62 quantification by itself does not reflect lysophagy. Immunofluorescence with LAMP1 was to assess the extent of overlap between lysosomes and α -Syn aggregates (Fig. 5a-d). However, immunofluorescence against p62 was performed to assess the overall impact of α -Syn aggregates on autophagy (Fig. 5h-k), and not lysophagy (as indicated in lines 335-341).

6) Neuronal differentiation and microglia differentiation with markers are not shown in the paper, please provide the characterization of iPSC differentiation with markers.

We thank the reviewer for this comment. Representative images of characterization of iPSC-derived neural progenitor cells (FOXA2+), mature neurons (MAP2+), dopaminergic neurons (TH+) and microglia (IBA1+, TMEM119+) have now been added in new Fig. S3k, l and updated in the text (lines 317-319).

7) TNTs were not labelled in the iPSC-derived neurons and microglia. It is important to know if IPSC differentiated cells also form TNTs similar to undifferentiated cell lines.

We thank the reviewer for this insightful comment. We fully agree that the propensity of iPSC-derived neurons and microglia to form TNTs is an important aspect to study. This is currently an ongoing collaborative study in the lab. Preliminary experiments demonstrated the presence of intercellular connections between both neurons, and microglia that could be morphologically identified as TNT-like connections (Figure R1). However, accurate quantification of these structures remains technically challenging due to several factors:

1. Differentiated neurons produce dense and overlapping neuritic networks, which complicates the discrimination between TNTs and neurites in standard-resolution confocal images.
2. Field-of-view limitations at high magnification (e.g., 40x) restrict the number of observable cell bodies, while lower magnifications (20x, 10x) limit the ability to resolve thin TNT-like structures.
3. In many fields, neurites are present without visible somas, making it difficult to definitively determine whether a connection bridges two distinct cells or originates from the same cell.

Given these limitations, quantifying TNTs in iPSC-derived cells in a robust and reproducible manner requires further protocol optimization. We are currently working on refining our imaging and analysis pipelines to enable reliable detection and quantification of TNTs in these more complex cellular environments.

Figure R1. Representative images of iPSC-derived neurons (DAn) and microglia (MG) stained with phalloidin to label TNT-like connections. Yellow arrowheads point towards intercellular connections. As evident, distinguishing the cells that are connected in case of neurons is a persistent challenge.

8) Line number 324, please check the figure number, would it be 6A instead of 5A?

We thank the reviewer for this comment and pointing out the mistake. We have now corrected it in the manuscript. The correct figure number is 6a.

Reviewer #2 (Remarks to the Author):

The manuscript by Chakraborty, Zurzolo and colleagues is very interesting, novel and generates new ideas in the field of autophagy and neurodegeneration. This study provides critical insights about how neurodegeneration-associated protein, which are transferred between cells via tunnelling nanotubes, could differentially affect autophagy in a cell-type specific manner. The authors elegantly demonstrate that microglial cells are more efficient in handling α -synuclein load and clearing them through autophagy, than the neuronal cells. Interestingly, they show that impairment of autophagy in neuronal cells causes transfer of α -synuclein to microglial cells for autophagic degradation. The study is undertaken by outstanding microscopy and relevant experimental systems to support the data. Overall, it is a beautiful story, and the manuscript is well written. The minor comments below will help to improve the manuscript that is already in good shape.

We thank the reviewer for their comments on this manuscript, which have helped us to improve the study and further strengthen our conclusions regarding the differential impact of α -Syn aggregates on neuronal and microglial lysosomes, and ascertain a protective role of microglia in clearing neuronal cell-derived aggregates. We have added new experiments and analyses as outlined in the response below.

[1] Although lysosomal functionality is shown in Fig. 2, it would be nice to show additional data on lysosomal pH using LysoSensor and LysoTracker in neuronal (SH-SY5Y) and microglial (HMC3) cells.

We thank the reviewer for this valuable suggestion. Following their advice we have performed new experiments using the pH-sensitive probe LysoSensor Green to directly assess lysosomal acidity in SH-SY5Y neuronal cells and HMC3 microglia.

These new results are now included in **Figure S1a–c**. Consistent with our earlier DQ-BSA analysis, we observed a marked shift toward neutral pH in neuronal lysosomes, while only a slight change was detected in microglial cells. These findings further support our conclusion that neuronal lysosomes are more significantly affected by α -Syn aggregates compared to microglial lysosomes.

We have updated the Results section accordingly (**lines 153–157**).

[2] Although greater nuclear localization of TFEB is shown after α -synuclein treatment in both neuronal and microglial cells (albeit at different levels) in Fig. 3, increase in TFEB target gene expression only occurs in microglial cells. Explain why this is not seen in neuronal cells. Would be good to check mTOR activation (using S6 or S6 kinase phosphorylation) and TFEB phosphorylation by immunoblotting in both these cell-types in the presence or absence of α -synuclein.

We thank the reviewer for raising this interesting point. To address the question regarding TFEB phosphorylation, we have performed immunoblotting to assess the

levels of S211-phospho-TFEB. Upon α -syn treatment we observed a ~34% reduction in neuronal cells, and ~29% reduction in microglia. This is in accordance with our previous immunofluorescence results of ~20% increase in nuclear TFEB localization in both the cell types. Additionally, to test mTOR activity, we have performed immunoblot assessment of the levels of S235/236-phospho-S6, and observed robust reductions in the extent of phosphorylation upon both α -Syn and rapamycin (positive control) treatments. Together, these results suggest an activation of autophagy in both cell types. These results have now been incorporated as new Fig. S3a-f and lines 237-244 in the main text.

Despite comparable nuclear TFEB translocation and upstream pathway activation there remains a cell-type difference in the downstream response – while microglial cells upregulate the expression of lysosomal genes, SQSTM1, and autophagy flux, neuronal cells fail to do so. We hypothesize such differential response may arise from cell-type-specific chromatin accessibility or transcriptional regulation that limits TFEB activity in neurons. This possibility has been discussed in the discussion section of the updated manuscript (lines 543-555).

[3] Fig. 4A: The LC3 immunoblotting data in bafilomycin clamp assay show inhibition of autophagosome formation in neuronal cells after α -synuclein treatment – which is correctly interpreted. However, the data in microglial cells might suggest an increase in autophagosome formation after α -synuclein treatment – although the authors state no change. Ideally, the quantification of LC3-II vs GAPDH in the Baf-treated conditions should be done in low exposure blots to capture any significant changes in LC3-II levels (to avoid quantifying overexposed bands due to autophagosome accumulation by Baf); whilst higher exposure blots can be used to quantify changes in no-Baf condition. Therefore, the LC3 blots should be re-quantified as suggested.

We thank the reviewer for this valuable observation and suggestion. We have now performed new rounds of experiments and re-analysis of LC3B-II levels using both low and high exposure blots as recommended. Our new results are similar to our previous observations, but now more precisely quantified:

- In low exposure blots (Bafilomycin A1-treated conditions), we observed a ~51% reduction in LC3B-II in neuronal cells, and a ~14% reduction in microglia following α -synuclein treatment.
- In high exposure blots, the reductions were ~44% in neurons and ~13% in microglia.
- Additionally, we assessed LC3B lipidation (conversion from LC3B-I to LC3B-II) in non-Bafilomycin-treated samples (high exposure blot analysis), and observed a ~27% reduction in neurons and only a ~4% reduction in microglia.

Together, these results reinforce our conclusion that autophagosome formation is significantly impaired in neuronal cells, while in microglial cells it is largely preserved, with no robust increase upon α -synuclein treatment.

We have incorporated these updated blots and quantifications in Figure 4a–c and Figure S3g–j, and have updated the manuscript text accordingly (lines 262–277).

[4] Related to the p62 data in Fig. 4, the authors state that autophagic flux is upregulated in microglial cells after α -synuclein treatment. Based on the data presented, it is ideal to state that autophagic flux is not overtly perturbed in these cells. However, the original conclusion could remain (with some modification – like microglia are trying to upregulate autophagic flux for clearance) if the authors find increased autophagosome formation in Fig. 4A after re-quantification.

We thank the reviewer for this suggestion, which was very helpful. We have now rephrased the sentences. Lines 295-296 and 309-311.

[5] The authors state that neuronal cells affect autophagy in microglial cells via α -synuclein transfer. For this, re-quantification of Fig. 4A is important. Additionally, in Fig. 6F and 6K, the authors could show the fold change of p62 in plus or minus Baf conditions in wild-type and α -synuclein conditions – current data is indicative of less p62 fold change in α -synuclein condition, which fits with the hypothesis. Here, can you show the levels of α -synuclein after co-culture to further support the notion about clearance of neuronal α -synuclein in microglia?

We thank the reviewer for this comment. As requested, we have now added the fold differences in Fig. 6f and 6k. As rightly pointed out by the reviewer, p62 levels in microglia exhibited a lower fold increase (~2-fold) when co-cultured with α -synuclein–burdened neuronal cells (Fig. 6f), or when exposed to their conditioned media (Fig. 6k), consistent with a bystander effect. In contrast, direct exposure of microglia to α -synuclein aggregates led to a ~3-fold increase in p62 levels, indicative of a cell-autonomous autophagic response.

Additionally, to directly support the notion that microglia participate in clearing neuronal α -synuclein, we have now quantified the number of α -Syn particles, and their overall load (number of α -Syn aggregates per unit area of cell) in neuronal cells when grown in mono-cultures versus co-cultures. In accordance with the observations that neuronal cells are efficient in trafficking aggregates to microglial cells, we observed a significantly reduced α -synuclein burden, both in terms of aggregate count and overall intensity per unit area, in neuronal cells when co-cultured with microglia. These graphs have now been added in Fig. 7d and 7e (new panels). These findings, together with the observed upregulation of microglial autophagy, support our hypothesis that microglia contribute to neuronal proteostasis by clearing transferred α -synuclein aggregates. The corresponding text in the Results section has been updated (lines 467–471).

[6] A summary cartoon of the key findings arising from this study would be ideal in Fig. 7.

We thank the reviewer for this suggestion. A schematic representation of the major findings of this study has now been added in Fig. 7, new panel f.

[7] Add information about LLOMe.

We thank the reviewer for this comment. Mechanism of LLOMe action has now been added in the manuscript text, lines 218-220.

[8] Formatting: Word spacing is missing in few instances.

We thank the reviewer for their observations, we have now fixed the mistakes.

References:

1. Henderson, J. M. *et al.* Tunnelling nanotube formation is driven by Eps8/IRSp53-dependent linear actin polymerization. *The EMBO Journal* 42, e113761 (2023).
2. Delage, E. *et al.* Differential identity of Filopodia and Tunneling Nanotubes revealed by the opposite functions of actin regulatory complexes. *Sci Rep* 6, 39632 (2016).
3. Hanna, S. J. *et al.* The Role of Rho-GTPases and actin polymerization during Macrophage Tunneling Nanotube Biogenesis. *Sci Rep* 7, 8547 (2017).
4. Drew, J. *et al.* Control of microglial dynamics by Arp2/3 and the autism and schizophrenia-associated protein Cyfip1. 2020.05.31.124941 Preprint at <https://doi.org/10.1101/2020.05.31.124941> (2020).
5. Paulson, S. G., Swafford, I., Lischka, F. W. & Rotty, J. D. The Arp2/3 complex is required for in situ haptotactic response of microglia to iC3b. 2025.05.21.655384 Preprint at <https://doi.org/10.1101/2025.05.21.655384> (2025).
6. Weichsel, J., Herold, N., Lehmann, M. J., Kräusslich, H.-G. & Schwarz, U. S. A quantitative measure for alterations in the actin cytoskeleton investigated with automated high-throughput microscopy. *Cytometry Part A* 77A, 52–63 (2010).
7. Miyano, T., Suzuki, A. & Sakamoto, N. Actin cytoskeletal reorganization is involved in hyperosmotic stress-induced autophagy in tubular epithelial cells. *Biochemical and Biophysical Research Communications* 663, 1–7 (2023).

I would like to thank the authors for their earnest effort for trying to improve the manuscript's quality.

We thank the Reviewer for the positive feedback and for the thoughtful comments and suggestions. In response, we performed some control experiments for this Reviewer, incorporated additional data in the manuscript and made several revisions throughout the manuscript to further strengthen its clarity and scientific rigor.

1. I began reviewing the revised manuscript and the rebuttal with high expectations, as it was from a top-tier TNT lab. However, I regret to say that the revised manuscript left me disheartened.

The authors tried to address the mechanistic link between autophagy impairment and TNT formation in neuronal and microglial cell lines. The authors added Supplementary Figure 8, where they have shown immunoblots of ARP3 and CDC42 in neuronal cells and microglia upon treatment with the autophagy inhibitor wortmannin (Wm) and flux blocker Baf A1. The authors claimed that in the figure Supplementary Figure 8a of WB, ARP3 and CDC42 in neuronal cells inhibited almost 50% (quantification in the Supplementary Figure 8b) upon autophagy inhibition by wortmannin (Wm) and flux blocker Baf A1. Loading control GAPDH of SH-SY5Y cells in the Wm-treated sample is visibly lower, and the normalized ARP3 and CDC42 levels cannot be around 50% as shown in the graph (Supplementary Figure 8b). In the represented WB ARP3 and CDC42 levels in Baf A1-treated sample show no reduction in band intensity compared to the control. None of the points in the quantitative graph, Supplementary Figure 8b, represents the WB figure of Supplementary Figure 8a. The mechanistic figure is the most important result to establish the causative role of autophagy inhibition in neuronal cells in the formation of TNTs towards microglia. However, authors represented the figure in the supplementary material, and the WB is not represented like other WBs of the manuscript with a Mw ladder, like Supplementary Figure 3a and d.

We thank the Reviewer for this detailed assessment and for carefully identifying the issue in the originally submitted western blot. The Reviewer is correct that the Arp2/3 blot for SH-SY5Y cells was not representative of the quantitation. We realized that it had been inadvertently mis-selected from the microglia blots; we have now replaced it with the correct representative blot in the revised Supplementary Fig. 9. Regarding CDC42, while our quantification showed a decreasing trend in SH-SY5Y cells upon autophagy inhibition, this effect did not reach statistical significance. We now state this explicitly in the Results section (see Page 9, Lines 430-433) to avoid any ambiguity. We would like to emphasize that although CDC42 is a central regulator of TNT biogenesis, its effect is context- and cell type-dependent, with positive and negative role reported (e.g. *Hanna et al Sci Rep 2017; Delage et al. Sci Rep 2016*). Overall, we believe that the strongest evidence comes from the Arp2/3 results that confirm our previously published results obtained in both SH-SY5Y cells and CAD neuronal cells (*Sartori-Rupp et al, Nat Comm 2019; Henderson et al EMBO J, 2023*). We also wish to underscore that these data, together with our observations on actin remodeling following autophagy inhibition, are intended to motivate further study rather than provide definitive conclusions; for this reason, they remain in the supplementary figures (see also comment 3 below).

Following the Reviewer's recommendation, we have also updated all Western blot images throughout the manuscript and supplementary materials to include molecular weight markers for consistency and transparency (revised Fig. 4, Supplementary Fig. 3 and Supplementary Fig. 9).

2. It is also not clear how Wortmannin (Wm) and flux blocker Baf A1, both can similarly affect ARP3 and CDC42 levels, where both Wm and Baf A1 pathways work with completely different molecular players. Authors have shown a change in actin fibres directional orientation with autophagy inhibition, both in SHSY5Y cells and microglia cells; however, both cells show similar effects. There are no differences between the two cell types.

We thank the Reviewer for raising this important point. We agree that wortmannin and bafilomycin A1 act through distinct upstream mechanisms, and we do not intend to imply that they converge on identical molecular players. Rather, our data indicate that autophagy blockade – regardless of whether it occurs at the initiation or late-lysosomal stage – results in a convergent cellular response that impacts actin organization.

In both SH-SY5Y and microglial cells, this manifests as increased actin coherence and altered stress fiber distribution. While the two cell types differ in lineage and function, the global actin rearrangements we report reflect a shared response to autophagy disruption, not necessarily identical upstream pathways. As the Reviewer notes, actin dynamics are highly complex and tightly regulated; context-dependent behaviors of actin-modifying proteins (including Arp2/3, CDC42, and additional regulators) are well-documented across different cell types.

Elucidating the precise molecular cascade that links autophagy impairment to specific actin regulators – and subsequently to TNT formation – would require an extensive mechanistic investigation that is beyond the scope of the present study. We now explicitly clarify this point in the revised manuscript (pages 13-14, lines 647-667). For this reason, and following the Reviewer's critique, we continue to present the actin remodeling analyses and Arp2/3/CDC42 data in the Supplementary Material, where they provide supportive evidence without overstating the mechanistic conclusions of the study.

3. Authors further disagreed in the rebuttal point 3, they have given the following points to argue that correlation of TNT with autophagy is not an indirect correlation. The authors claimed direct correlation, stating that neuronal autophagy failure triggers the directional transfer of α -syn aggregates to microglia via TNTs as an alternative clearance route. However, the stated logics are weak. Please see the argument below.

- Autophagy inhibition in neurons (using wortmannin and bafilomycin A1) significantly increases TNT formation. Autophagy inhibition increases aggregation/aberrant organelle load in neurons, which is known to increase oxidative stress and ROS. Several studies have shown that oxidative stress and ROS play a crucial role in TNT formation. Thus, the above data do not prove a direct correlation between autophagy and TNTs.

- At the molecular level, reduced levels of ARP3 and CDC42 and actin cytoskeletal reorganization (as shown in response to previous comments) are consistent with a cellular state favoring TNT formation. The reviewer is sorry to state that the WB data are not solid, and this is explained above.

- Our experimental data indicate that secreted α -Syn contributes minimally to microglial aggregate load under these conditions, whereas direct transfer via TNTs is the dominant mechanism.

Increased accumulation of aggregates induces TNTs and TNT-mediated cell-to-cell transfer in many cell types. Several studies have shown that, and the lead author of this manuscript has earlier written a wonderful review on that in Victoria et al JCB 2017. This more reflects a direct correlation between the accumulation of aggregates with TNT-mediated transfer, not the autophagy inhibition with TNT formation.

We thank the Reviewer for this insightful comment.

Firstly, we fully agree that autophagy inhibition leads to increased protein and organellar stress, which elevates oxidative stress and ROS levels – factors known to promote TNT formation in various cell types. To briefly recapitulate our findings:

- 1- α -Syn treatment increases TNTs, which are then hijacked by α -syn for spreading.
- 2- α -Syn burden combined to autophagic blockade further increases α -syn transfer.

To directly address the Reviewer's important question - whether the increase in α -syn transfer upon autophagy inhibition could be explained solely by an additional rise in ROS - we quantified ROS levels in both neuronal (SH-SY5Y) and microglial cells treated with α -syn alone or in combination with wortmannin or bafilomycin A1, using H_2O_2 as a positive control to confirm assay sensitivity. Importantly, ROS measurements were performed at the same time points used in our autophagy-inhibition experiments (2 h wortmannin; 4 h bafilomycin A1), ensuring direct comparability. This data is now added in Supplementary Fig. 10, and the manuscript has been updated in page 9, lines 437-444. In brief, while α -syn alone induces ROS accumulation, the autophagy blockade conditions used throughout our study did not increase ROS levels beyond those induced by α -syn treatment alone. Thus, although ROS may contribute to baseline TNT induction under α -syn exposure, the additional enhancement of α -syn transfer observed upon autophagy inhibition cannot be accounted for by further ROS elevation.

Secondly, regarding Arp2/3 western blot, we apologize for the mistake in the representative image; we have now replaced it with the correct blot. Please see our full reply above.

Thirdly, we thank the Reviewer for highlighting the important point of aggregate load in TNT-mediated transfer. To disentangle the contribution of aggregate burden from that of autophagy impairment in α -syn transfer, we performed several additional experiments:

First, we pharmacologically modulated autophagy in α -syn-burdened cells. Rapamycin treatment, which enhances autophagy, significantly reduced the number of α -syn aggregates per cell (data added in Supplementary Fig. 5e and f). Consistently, rapamycin-treated donor neurons exhibited reduced α -syn transfer to microglia in a contact-dependent manner (revised Fig. 6f-h). These results are now described in the revised

manuscript (page 8, lines 354-375). Importantly, the observed changes in α -syn transfer upon autophagy modulation could reflect either: (i) a correlation between aggregate load and transfer efficiency, and/or (ii) a direct effect of autophagy enhancement or inhibition on the mechanisms that govern α -syn propagation.

To address this question, we exposed neuronal cells to different α -syn aggregate doses (100 nM and 250 nM, compared with the 500 nM used previously) and quantified the extent of aggregate transfer. Under basal conditions, neither parameter changed across the three doses: 31–35% of acceptor cells were observed, with ~2.17–2.52 aggregates per cell (updated Supplementary Fig. 6). In contrast, autophagy inhibition with wortmannin increased α -syn transfer at all doses tested, reaching ~63–67% acceptor cells (Supplementary Fig. 6b and d) and ~3.33–3.34 aggregates per cell (Supplementary Fig. 6c and e). These findings indicate that impaired clearance promotes preferential disposal of aggregates to neighboring cells in a contact-dependent manner, irrespective of the initial α -syn load. The manuscript text has been updated in pages 8, lines 377-387 highlighting these results.

In addition, to determine whether this increased transfer reflects a more general effect of autophagic blockade rather than a phenomenon specific to α -syn, we used a DiD-based vesicle transfer assay. DiD labels intracellular vesicles; this approach has been widely used to evaluate intercellular communication. Wortmannin treatment significantly increased DiD+ vesicle transfer from neuronal cells to microglia in a contact-dependent manner (Supplementary Fig. 8c–g), demonstrating that autophagy inhibition enhances intracellular transfer capacity even when the cargo is not α -syn. This is now highlighted in the revised manuscript in page 9, lines 410-418.

We also acknowledge that long-term impairment of autophagy could, in principle, increase α -syn burden and thereby influence transfer indirectly. However, the short treatment paradigms used here – together with the α -syn dose-response experiments and the neutral DiD cargo assay – allow us to clearly separate the acute effects of autophagy inhibition from aggregate load itself. Within this controlled experimental context, the enhanced transfer cannot be attributed to differences in initial α -syn burden.

Taken together, these data clearly indicate that autophagic impairment independently enhances TNT-mediated material transfer. We observe both an increased percentage of recipient cells and a higher number of cargoes transferred, supporting the conclusion that autophagy inhibition directly promotes TNT-mediated propagation beyond its effects on aggregate burden.

We hope these additional analyses satisfactorily address the Reviewer's concerns and help clarify the mechanistic implications of our findings

4. TNTs between iPSC-derived neurons and microglia are not shown properly. The authors stated the following in rebuttal point 8. "Given these limitations, quantifying TNTs in iPSC-derived cells in a robust and reproducible manner requires further protocol optimization. We are currently working on

refining our imaging and analysis pipelines to enable reliable detection and quantification of TNTs in these more complex cellular environments.”

We appreciate the Reviewer’s concern regarding the visualization and quantification of TNTs in iPSC-derived neurons and microglia. We respectfully emphasize that our cautious interpretation reflects current challenges widely recognized in the field. The criteria for TNT identification are not consistently standardized across the literature, which has contributed to variability and confusion in the assessment.

By explicitly acknowledging these limitations, our study aims to promote a more rigorous and transparent framework for defining TNTs in complex iPSC-derived cellular models. The data presented in Fig. 5 are intended to highlight a conserved cellular response to α -syn aggregates in iPSC-derived neurons and microglia, consistent with our observations in established cell lines.

We are actively refining our imaging and analysis protocols to enable more robust and reproducible TNT quantification in these models, and we have clearly stated this ongoing effort in the revised manuscript.

Thank you for addressing the other questions. The reviewer is not convinced that neuronal autophagy failure is the cause that drives TNT-mediated α -Synuclein transfer to microglia to outsource aggregate clearance. This is a nice correlation study that shows accumulation of aggregates in the neuron drives TNT-mediated α -Synuclein transfer to microglia to outsource aggregate clearance. The authors established the correlation of aggregate accumulation with TNT beautifully. But the study is definitely not a study where one can say autophagy inhibition is the cause of TNT formation in neurons, and autophagy inhibition differentially induces TNTs in neurons and microglia.

We thank the Reviewer for his/her careful evaluation and for raising insightful points regarding the mechanisms linking autophagy impairment to TNT-mediated α -syn transfer. The additional experiments and clarifications included in the revised manuscript reinforce the conclusion that autophagy blockade directly enhances TNT-mediated material transfer. In particular, we demonstrate that:

1. The increase in α -syn transfer upon autophagy inhibition cannot be explained by ROS elevation.

Although α -syn treatment induces ROS in both neuronal and microglial cells, autophagy inhibition with wortmannin or bafilomycin A1 does not further increase ROS levels (as mentioned previously). Thus, ROS does not account for the enhanced transfer observed under autophagy-blocked conditions.

2. The enhanced transfer is independent of aggregate burden.

Reducing the initial α -syn load does not alter basal transfer, whereas autophagy inhibition increases transfer at all doses tested. Moreover, DiD-labeled vesicles also display increased transfer upon autophagy inhibition. These findings directly address the Reviewer’s concern and show that autophagy blockade promotes intercellular transfer beyond its effects on α -syn accumulation.

3. Corrected Arp2/3 data support an effect of autophagy impairment on actin-regulatory pathways. The representative Arp2/3 western blot has now been replaced with the correct version and is

fully consistent with the quantified trend. Establishing a definitive causal cascade would require an extensive mechanistic investigation that we believe lies beyond the scope of this study.

Taken together, these new data and clarifications strengthen the mechanistic framework of the study by ruling out alternative explanations (ROS, aggregate load) and by supporting a functional link between autophagy impairment and increased TNT-mediated α -syn transfer. We believe these points address the Reviewer's concerns and provide a clear basis for the conclusions presented in the revised manuscript.